# Rate of brain aging associates with future executive function in Asian children and older adults

Susan F Cheng[1,2], Wan Lin Yue[1,2], Kwun Kei Ng[2], Xing Qian[2], Siwei Liu[2], Trevor WK Tan[1,2], Kim-Ngan Nguyen[2], Ruth LF Leong[2], Saima Hilal[2,3,4], Ching-Yu Cheng[5,6], Ai Peng Tan[2,7,8,9], Evelyn C Law[7,9], Peter D Gluckman[7,10], Christopher Li-Hsian Chen[2,4], Yap Seng Chong[2,7,9], Michael J Meaney[2,7,11,12], Michael WL Chee[2], BT Thomas Yeo[1,2,13,14†], Juan Helen Zhou[1,2,13*†]

[1]Integrative Sciences and Engineering Programme, NUS Graduate School, National University of Singapore, Singapore, Singapore; [2]Yong Loo Lin School of Medicine, National University of Singapore, Singapore, Singapore; [3]Saw Swee Hock School of Public Health, National University of Singapore and National University Health System, Singapore, Singapore; [4]Memory Aging and Cognition Centre, National University Health System, Singapore, Singapore; [5]Singapore Eye Research Institute, Singapore National Eye Centre, Singapore, Singapore; [6]Duke-NUS Medical School, Singapore, Singapore; [7]Singapore Institute for Clinical Sciences (SICS), A*STAR Research Entities (ARES), Singapore, Singapore; [8]Brain–Body Initiative Program, Agency for Science, Technology and Research (A*STAR), Singapore, Singapore; [9]National University Health System, Singapore, Singapore; [10]Liggins Institute, University of Auckland, Auckland, New Zealand; [11]Douglas Mental Health University Institute, McGill University, Montreal, Canada; [12]Strategic Research Program, A*STAR Research Entities (ARES), Singapore, Singapore; [13]Department of Electrical and Computer Engineering, National University of Singapore, Singapore, Singapore; [14]N.1 Institute for Health & Institute for Digital Medicine (WisDM), National University of Singapore, Singapore, Singapore

*For correspondence:
helen.zhou@nus.edu.sg

†These authors contributed equally to this work

## eLife Assessment

This **valuable** study marks a significant advancement in brain aging research by centering on Asian populations (Chinese, Malay, and Indian Singaporeans), a group frequently underrepresented in such studies. It unveils **solid** evidence for anatomical differences in brain aging predictors between the young and old age groups. Overall, this study broadens our understanding of brain aging across diverse ethnicities.

**Abstract** Brain age has emerged as a powerful tool to understand neuroanatomical aging and its link to health outcomes like cognition. However, there remains a lack of studies investigating the rate of brain aging and its relationship to cognition. Furthermore, most brain age models are trained and tested on cross-sectional data from primarily Caucasian, adult participants. It is thus unclear how well these models generalize to non-Caucasian participants, especially children. Here, we tested a previously published deep learning model on Singaporean elderly participants (55–88 years old) and children (4–11 years old). We found that the model directly generalized to the elderly participants, but model finetuning was necessary for children. After finetuning, we found that the rate of change

in brain age gap was associated with future executive function performance in both elderly participants and children. We further found that lateral ventricles and frontal areas contributed to brain age prediction in elderly participants, while white matter and posterior brain regions were more important in predicting brain age of children. Taken together, our results suggest that there is potential for generalizing brain age models to diverse populations. Moreover, the longitudinal change in brain age gap reflects developing and aging processes in the brain, relating to future cognitive function.

## Introduction

The human brain undergoes coordinated, multidimensional anatomical changes throughout the lifespan, which can be measured noninvasively by MRI (*Bethlehem et al., 2022*). These anatomical changes occur in parallel with age-related changes to other measurable phenotypes, such as cognition (*Hedden and Gabrieli, 2004*; *Roalf et al., 2014*). Abnormal aging in late life and abnormal development in early life have both been implicated with increased risk of neuropsychiatric disorders (*van der Flier and Scheltens, 2005*; *Marsh et al., 2008*). Thus, efforts have been made to quantify the heterogenous effects of aging/development on the brain through the concept of 'brain age'. Brain age uses machine learning to predict age from neuroimaging data. A higher brain age suggests more advanced aging/development relative to one's chronological age. This helps summarize complex patterns into a single number that preserves individual variations (*Franke et al., 2012*).

Historically, most brain age studies first use specialized software to preprocess MRI data and extract features such as gray matter volume, cortical thickness, or surface area. A machine learning model is then trained to predict age from the extracted features. The model is typically trained using cognitively normal participants, with chronological age acting as the ground truth. The model is then applied to new participants to predict their brain age (*Gaser et al., 2013*; *Cole et al., 2018*; *Kaufmann et al., 2019*; *Tian et al., 2023*).

More recently, deep learning models have gained popularity over traditional machine learning models for brain age prediction (*Wood et al., 2022*; *Hofmann et al., 2022*; *Yin et al., 2023*; *Leonardsen et al., 2022*; *Bashyam et al., 2020*; *Chen et al., 2020*). Unlike previous machine learning methods, deep learning models can learn relevant features directly from the unprocessed (or minimally processed) MRI scan. This reduces the need for specialized preprocessing to extract features, which is time-consuming, requires expert knowledge, and involves laborious quality control. This also allows deep learning models to train on increasingly numerous and heterogeneous data. Some of these pretrained models have been made publicly available and shown impressive generalization performance on completely unseen test data (*Leonardsen et al., 2022*; *Bashyam et al., 2020*). However, both training and testing data still primarily consist of Caucasian participants, which could bias the models (*Hahn et al., 2022*; *Li et al., 2022*). Deep learning models can use finetuning (i.e., transfer learning) to help overcome this bias and achieve good performance even in small datasets (*Chen et al., 2020*). But to our knowledge, previous work has not examined the performance, with and without finetuning, of a Caucasian-centric model on non-Caucasian children and elderly participants, such as Asian children and elderly participants. This generalization is important to establish as the prevalence of developmental disorders (*Qiu et al., 2020*) and dementia (*Catindig et al., 2012*) are on the rise in Asia.

In addition to being able to predict age, predictions made by the model should show utility in relating to other phenotypes of interest (*Jirsaraie et al., 2023a*). The deviation from expected aging is often quantified as the brain age gap (BAG; also referred to as brainPAD, brainAGE, etc.), which is calculated by subtracting chronological age from brain age. The BAG has shown broad associations with brain disorders (*Kaufmann et al., 2019*), risk of mortality (*Cole et al., 2018*), and cognitive function (*Wrigglesworth et al., 2022*), to name a few (see *Wrigglesworth et al., 2021*; *Baecker et al., 2021* for recent reviews). However, there are relatively few longitudinal studies in healthy participants. Previous work found associations with early life or cross-sectional measures (*Wrigglesworth et al., 2022*; *Vidal-Pineiro et al., 2021*; *Elliott et al., 2021*; *Franke et al., 2018*), but there was only a weak link to future age-related cognitive decline, which did not survive multiple comparison correction (*Wrigglesworth et al., 2022*). Notably, these studies only used brain age measured at one time point. There is evidence that cross-sectional and longitudinal brain measures may reflect different

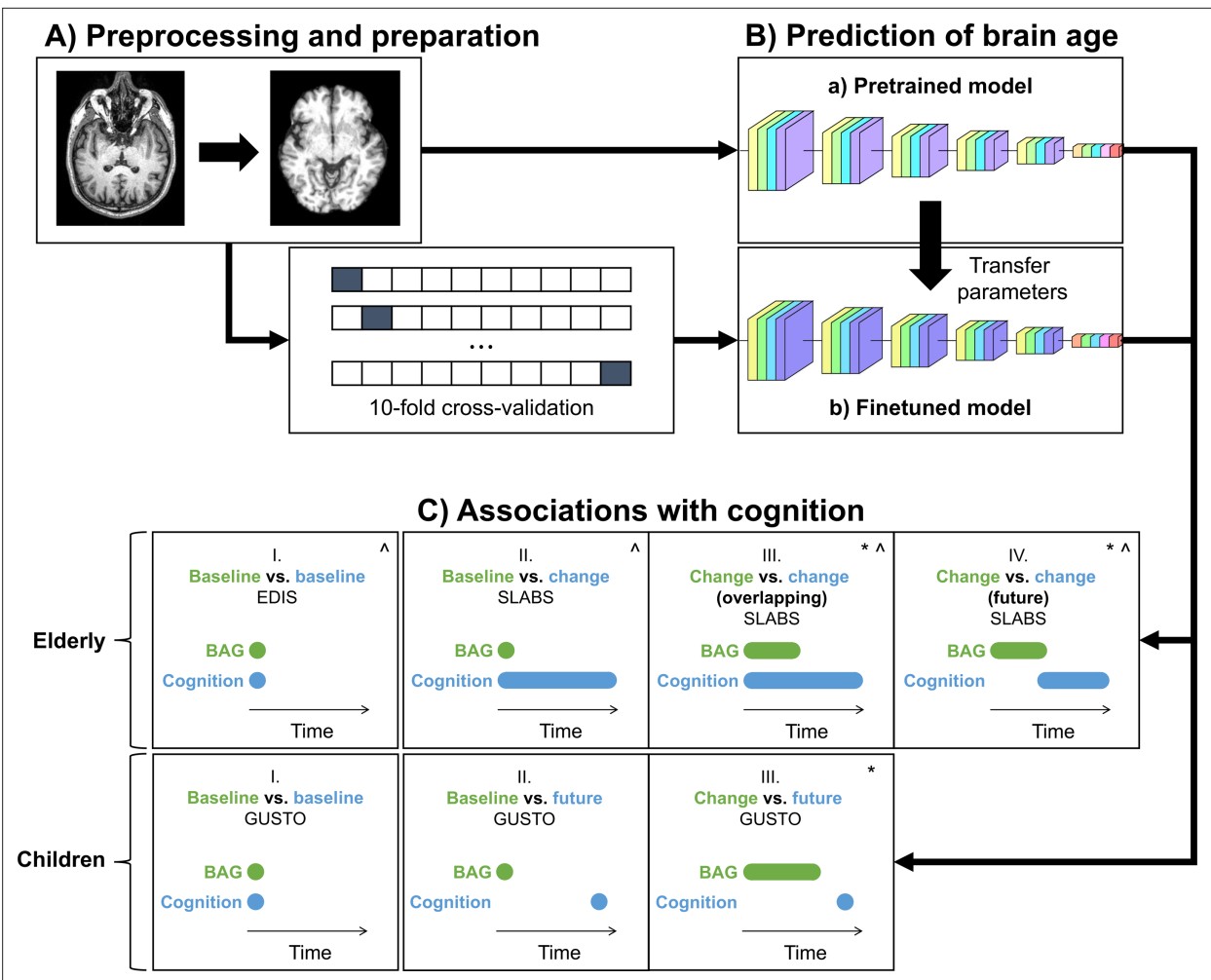

**Figure 1.** Study design schematic. (**A, B**) T1 MRI scans were minimally preprocessed according to the simple fully convolutional network (SFCN) pipeline (*Leonardsen et al., 2022*). These were (a) directly input into the pretrained brain age model or (b) split into 10 cross-validation folds to finetune the model. The finetuned model transferred the weights from the pretrained model for initialization. All layers were then retrained. Age predictions were obtained on the test folds. BAG was calculated by subtracting chronological age from predicted age. Model interpretability was interrogated using guided backpropagation. (**C**) Cross-sectional and longitudinal association of BAG and cognitive performance were tested using multiple linear regression models in both elderly and children. Time intervals for BAG and cognition, based on data availability, are shown schematically. Annual rate of change was calculated from a linear regression with time for each participant. All models included chronological age and sex as covariates.:^ models for elderly also included years of education as a covariate;* models with (annual rate of) change in BAG also included baseline BAG as a covariate. EDIS, Epidemiology of Dementia in Singapore; SLABS, Singapore-Longitudinal Aging Brain Study; GUSTO, Growing Up in Singapore Towards healthy Outcomes; BAG, brain age gap.

factors and predispositions of individuals (*Walhovd et al., 2023a*), suggesting that combining them could provide more predictive power. However, to our knowledge, the additional utility of longitudinal changes in brain age has not been tested in healthy participants.

Thus, in this work, we leverage a state-of-the-art deep learning brain age model trained on over 30,000 individuals across the lifespan (*Leonardsen et al., 2022*) to test generalizability to Asian elderly participants and children. We also finetune the model to explore how much predictions improve. We then examine the longitudinal utility of brain age in associating with future cognition. Finally, we investigate model interpretability using guided backpropagation. Our findings provide insight into the generalizability of brain age models and the importance of longitudinal measurements.

**Table 1.** Participant characteristics at baseline.

EDIS was cross-sectional, while SLABS and GUSTO were longitudinal. Reported as mean ± standard deviation (range). *GUSTO ethnicities were based on the mother. M/F, male/female; C/M/I/O, Chinese/Malay/Indian/Other; MMSE, Mini-Mental State Examination; EDIS, Epidemiology of Dementia in Singapore; SLABS, Singapore Longitudinal Aging Brain Study; GUSTO, Growing Up in Singapore Towards healthy Outcomes.

| | Elderly | | Children |
|---|---|---|---|
| Characteristic | EDIS (N=694) | SLABS (N=215) | GUSTO (N=678) |
| Age (years) | 69.91 ± 6.46 (60−88) | 68.17 ± 6.77 (55−85) | 5.85 ± 1.68 (4.2−11.3) |
| Sex (M/F) | 340/354 | 101/114 | 346/332 |
| Ethnicity (C/M/I/O) | 276/184/234/0 | 215/0/0/0 | 370/187/120/1* |
| Education (years) | 6.18 ± 4.63 (0−22) | 12.02 ± 3.45 (0−21) | N/A |
| MMSE score | 24.13 ± 3.59 (10−30) | 28.29 ± 1.27 (26−30) | N/A |
| Imaging follow-up (years) | N/A | 4.00 ± 3.33 (0−9.59) | 3.49 ± 2.41 (0−6.69) |
| Cognition sample size | N=694 | N=81–212 | N=217–239 |

## Results

*Figure 1* shows the study design. For the brain age model, we used the publicly available simple fully convolutional network (SFCN) pretrained on 34,285 T1 MRI scans from 21 non-overlapping datasets across the lifespan (*Leonardsen et al., 2022*). This pretrained model (a.k.a. pyment) was found to have the highest accuracy and test−retest reliability in a recent comparison of publicly available brain age models (*Dörfel et al., 2023*). While featuring an unusually large and heterogenous training set for brain age prediction, there was still a relative lack of training data from very young and/or non-Caucasian participants.

Thus, to test generalizability to Asian elderly participants and children, we used three datasets from Singapore: (1) the cross-sectional Epidemiology of Dementia in Singapore (EDIS) study (*Hilal et al., 2013*; *Hilal et al., 2017*; *Wong et al., 2019*), consisting of 694 non-demented elderly (226 with no cognitive impairment [NCI] and 468 with cognitive impairment no dementia[CIND]); (2) the longitudinal Singapore Longitudinal Aging Brain Study (SLABS) (*Chee et al., 2009*), consisting of 215 healthy elderly participants; and (3) the longitudinal Growing Up in Singapore Towards healthy Outcomes (GUSTO) study (*Soh et al., 2014*), consisting of 678 healthy children. These datasets are detailed in 'Materials and methods'. *Table 1* and *Appendix 2—table 1* summarize the participant demographic and cognitive characteristics.

### The pretrained brain age model performs well in older adults, while the finetuned model performs well in both older adults and children

We first input minimally preprocessed T1 scans directly to the pretrained model (all baseline and follow-up data). We also finetuned the model for our local datasets using 10-fold cross-validation (*Figure 1A and B*; section 'Brain age predictions'). *Figure 2* shows the brain age predictions for the pretrained and finetuned models on all datasets.

In EDIS and SLABS (elderly), the pretrained model performed well, as evidenced by the high correlation ($r = 0.7389$ for EDIS and $r = 0.8136$ for SLABS) and low MAE (MAE = 3.9895 for EDIS and MAE = 3.4668 for SLABS; *Figure 2A*, first two rows). After finetuning, correlations and MAEs slightly improved ($r = 0.7445$ for EDIS and $r = 0.8138$ for SLABS; MAE = 3.3232 for EDIS and MAE = 3.2653 for SLABS; *Figure 2B*, first two rows), but the predictions were generally similar to those made by the pretrained model (correlation between finetuned and pretrained predictions = 0.9143 for EDIS and 0.9231 for SLABS).

In contrast, the pretrained model did not perform as well in GUSTO (children). The MAE was lower than in elderly (MAE = 2.57), but the age range of GUSTO was also much smaller. Importantly, predictions did not distinguish among younger ages, leading to a low correlation ($r = 0.5426$; *Figure 2A*, last row). After finetuning, the correlation and MAE drastically improved ($r = 0.9411$; MAE = 0.6286; *Figure 2B*, last row). The variance in predicted ages also increased as chronological age increased

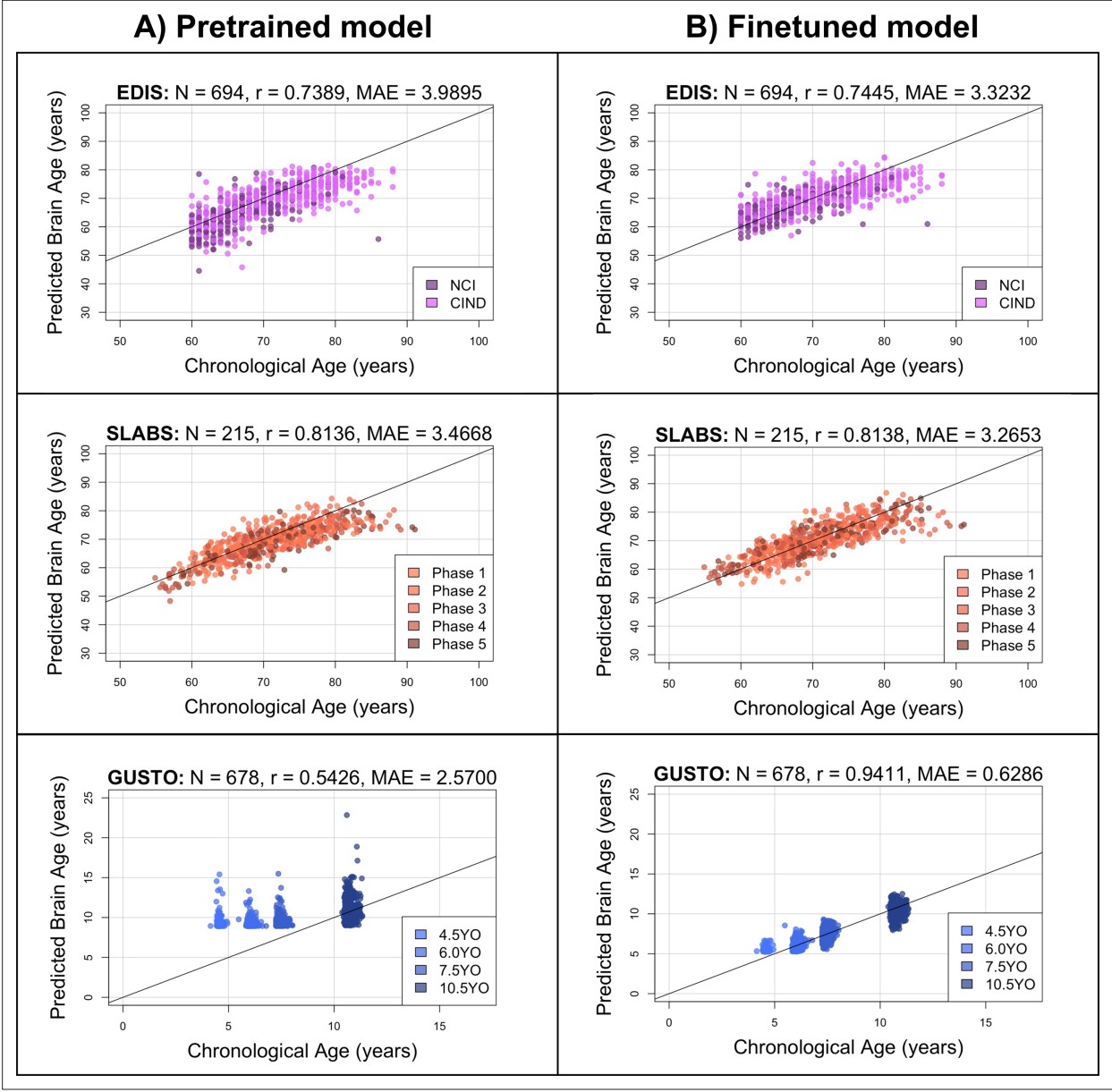

**Figure 2.** The pretrained brain age model performs well in elderly participants, while the finetuned model performs well in both elderly participants and children. Black identity lines representing perfect prediction are included for reference. (**A**) Predicted brain ages from the pretrained model are plotted against chronological age. They are highly correlated for EDIS and SLABS (elderly), but not GUSTO (children). (**B**) Predicted brain ages from the finetuned model are plotted against chronological age. They are highly correlated in all three datasets. EDIS, Epidemiology of Dementia in Singapore; SLABS, Singapore-Longitudinal Aging Brain Study; GUSTO, Growing Up in Singapore Towards healthy Outcomes; N, number of participants; r, Pearson's correlation coefficient; MAE, mean absolute error; NCI, no cognitive impairment; CIND, cognitive impairment no dementia.

The online version of this article includes the following figure supplement(s) for figure 2:

**Figure supplement 1.** Variance of finetuned predicted ages by age group in Growing Up in Singapore Towards healthy Outcomes GUSTO.

(*Figure 2—figure supplement 1*). Unlike EDIS and SLABS, the finetuned predictions were not similar to the pretrained predictions (correlation = 0.5732).

## The brain age gap is negatively associated with executive function in elderly participants

To validate the brain age model with a large and cognitively heterogeneous sample, we first tested cross-sectional associations in EDIS (N=694) using multiple linear regression models (*Figure 1C*).

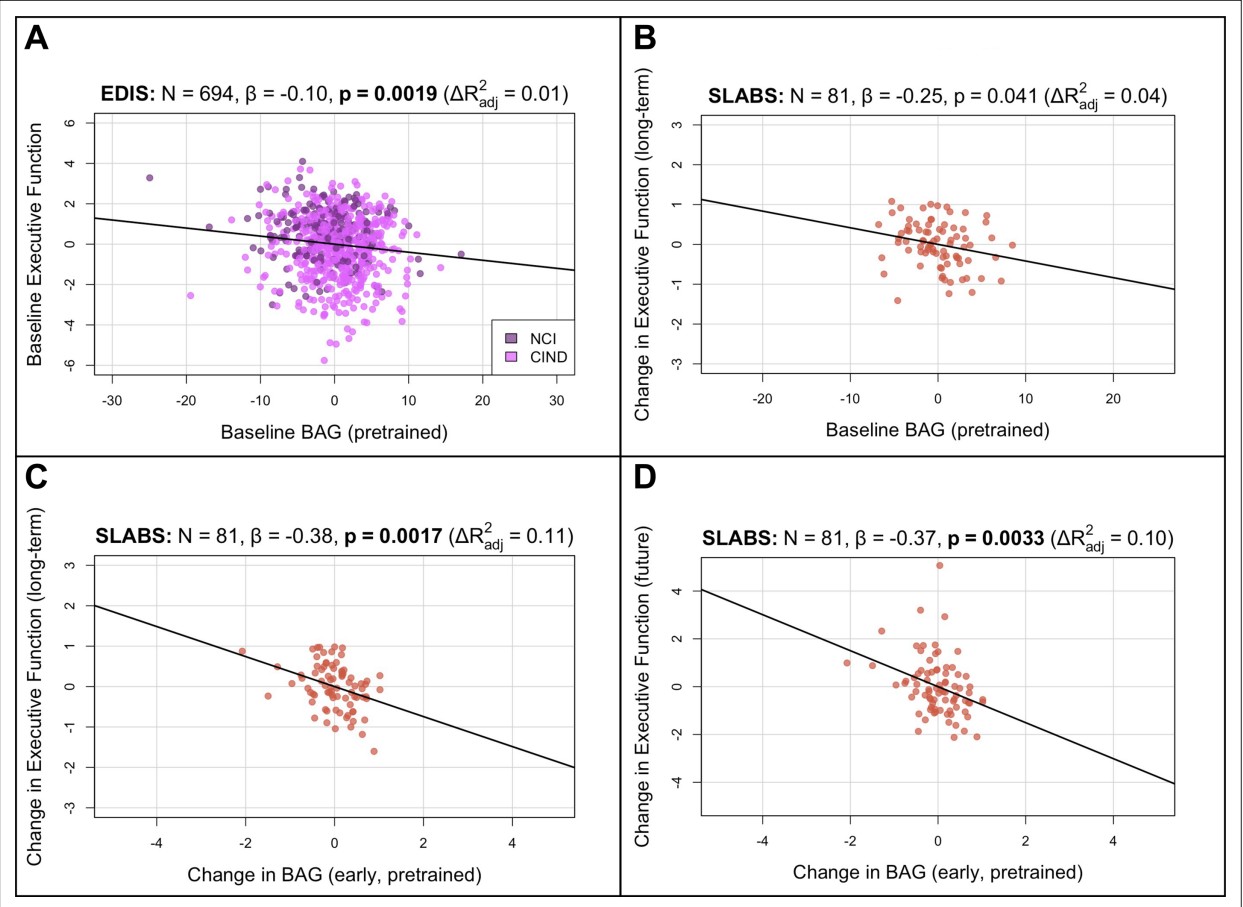

**Figure 3.** BAG from the pretrained model is negatively associated with executive function in elderly participants. Bolded p-values indicate significance after Holm-Bonferroni correction ($p_{corr} < 0.05$). All models include chronological age, sex, and years of education as covariates. Models with change in BAG also include baseline BAG as a covariate. Results are similar after finetuning (*Figure 3—figure supplement 1*). (**A**) Partial regression plot between baseline BAG and executive function in EDIS, colored by cognitive status. A significant negative association is observed. (**B**) Partial regression plot between baseline BAG and long-term rate of change in executive function (mean follow-up time = years) in SLABS. A negative association is observed, but it is not significant after correcting for multiple comparisons. (**C**) Partial regression plot of early longitudinal rate of change in BAG (mean follow-up time = years) when added to the model in (**B**). A significant negative association and increase in $R^2$ is observed. (**D**) Partial regression plot as in (**C**), but with future rate of change in executive function (mean follow-up time = years), removing the overlap with early change in BAG. A significant negative association is again observed. N, number of participants; β, standardized regression coefficient; p, p-value for variable of interest (x-axis); $\Delta R^2_{adj}$, change in adjusted $R^2$ when adding variable of interest; BAG, brain age gap; NCI, no cognitive impairment; CIND, cognitive impairment no dementia; EDIS, Epidemiology of Dementia in Singapore; SLABS – Singapore-Longitudinal Aging Brain Study.

The online version of this article includes the following figure supplement(s) for figure 3:

**Figure supplement 1.** Brain age gap from the finetuned model remains negatively associated with executive function in elderly.

We included chronological age, sex, and years of education as covariates. Higher baseline BAG was broadly associated with lower baseline cognitive performance (i.e., negative associations; *Appendix 3—table 1*). The associations were significant after multiple comparison correction for global cognition ($\beta = -0.1125$, $p_{corr} = 0.0006$), executive function ($\beta = -0.1029$, $p_{corr} = 0.0076$, *Figure 3A*), language ($\beta = -0.1145$, $p_{corr} = 0.0047$), visuomotor speed ($\beta = -0.0825$, $p_{corr} = 0.0136$), visuoconstruction ($\beta = -0.0896$, $p_{corr} = 0.0136$), verbal memory ($\beta = -0.1096$, $p_{corr} = 0.0034$), and visual memory ($\beta = -0.1395$, $p_{corr} = 0.0002$). The association was not significant for attention ($\beta = -0.0404$, $p_{corr} = 0.2461$). These results were consistent after finetuning (*Figure 3—figure supplement 1A* and *Appendix 3—table 1*). Similar broad negative associations were also observed in SLABS at baseline (N=212), but these were not significant (*Appendix 3—tables 3 and table 4*).

To investigate longitudinal utility of brain age in healthy elderly, we tested associations in a longitudinal subset of SLABS (N=81) using similar multiple linear regression models. We first related baseline

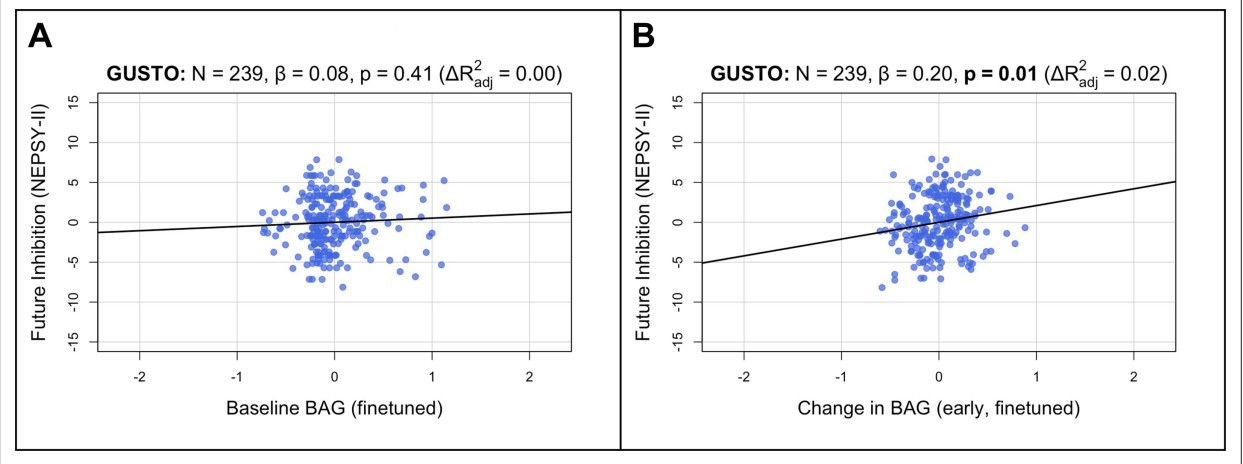

**Figure 4.** Longitudinal BAG from the finetuned model is positively associated with inhibition in children. Bolded p-values indicate significance after Holm–Bonferroni correction ($p_{corr} < 0.05$). All models include chronological age and sex as covariates. Models with change in BAG also include baseline BAG as a covariate. (**A**) Partial regression plot between baseline BAG (calculated from 4.5 or 6.0 years old) and future NEPSY-II inhibition scaled subscore (measured at 8.5 years old). No significant association is observed. (**B**) Partial regression plot of early longitudinal rate of change in BAG calculated from 4.5 to 7.5 years old (mean follow-up time = 2.4 ± 0.7 years) when added to the model in (**A**). A significant positive association and increase in $R^2$ is observed. N, number of participants; β, standardized regression coefficient; p, p-value for variable of interest (x-axis); $\Delta R^2_{adj}$, change in adjusted $R^2$ when adding variable of interest; BAG, brain age gap; GUSTO, Growing Up in Singapore Towards healthy Outcomes.

The online version of this article includes the following figure supplement(s) for figure 4:

**Figure supplement 1.** Brain age gap (BAG) from the pretrained model is not associated with inhibition in children.

**Figure supplement 2.** Baseline BAG is not associated with baseline IQ in children.

BAG and early change in BAG to long-term cognitive change (*Figure 1C*; section 'Associations with cognition'). Baseline BAG generally failed to show associations with longitudinal cognitive changes (*Appendix 3—table 5*). While higher baseline BAG was associated with faster long-term decline in executive function, it was not significant after multiple comparison correction (β =−0.2477, $p = 0.0406$, $p_{corr} = 0.2433$, *Figure 3B*). On the other hand, the early rate of BAG change was negatively associated with long-term rate of executive function change (β =−0.3807, $p = 0.0017$, $p_{corr} = 0.0100$, *Figure 3C*). This negative association held after removing the temporal overlap between BAG and cognition, looking only at the future rate of executive function change (β =−0.3807, $p = 0.0033$, *Figure 3D*). Notably, these associations were independent of baseline BAG, chronological age, sex, and years of education. The associations were also specific to executive function (*Appendix 3—table 7*). Results were again consistent after finetuning (*Figure 3—figure supplement 1B–D*, *Appendix 3—tables 6* and *8*).

## The longitudinal change in brain age gap is positively associated with inhibition in children

To extend our analyses to healthy children, we tested cross-sectional and longitudinal associations in GUSTO using multiple linear regression models similar to above (*Figure 1C*; section 'Associations with cognition'). Since longitudinal cognitive data was not available for GUSTO, we used the cognitive scores themselves instead of the change. Furthermore, since finetuning the model drastically improved prediction accuracy in GUSTO, we used the finetuned predictions for our main analyses. We did not find a significant association between baseline BAG and baseline IQ score (β=−0.0618, $p = 0.3809$, *Figure 4—figure supplement 2B*). Similarly, we did not find significant associations between baseline BAG and future cognitive scores (at 8.5 years old; $|β| \leq 0.0829$ $p \geq 0.4086$, *Figure 4A*; *Appendix 3—table 10*). However, the early rate of BAG change (from 4.5 to 7.5 years old) was positively associated with future inhibition (at 8.5 years old; $β = 0.2006$, $p = 0.0103, p_{corr} = 0.0411$, *Figure 4B*). The early rate of BAG change was also positively associated with future switching, but it was not significant after correcting for multiple comparisons ($β = 0.1795$, $p = 0.0221$, $p_{corr} = 0.0663$, *Appendix 3—table 12*). These associations were independent of baseline BAG, chronological age, and sex. Notably,

in contrast to older adults, the direction of association was now positive, meaning increased early rate of BAG change was associated with better future executive function performance. There were no significant associations using the pretrained model (*Figure 4—figure supplements 1 and 2A*, *Appendix 3—tables 9* and *11*).

## Finetuned brain age models focus on distinct features in children and elderly participants

Finally, to investigate model interpretability, we used guided backpropagation (*Springenberg et al., 2015*) to derive group-level saliency maps for brain age prediction (section 'Model interpretability'). *Figure 5* shows the top 10% of contributing voxels to age prediction in four representative slices (left) for all datasets. Full 3D maps are also made available online. Relative contributions of white and gray matter features across the whole brain are shown on the right. Areas near the lateral ventricles are labeled in red, while areas more prominent in elderly than children are labeled in magenta, and areas more prominent in children than elderly are labeled in blue.

Both EDIS and SLABS show similar profiles (*Figure 5A and B*), suggesting important features are stable across the elderly datasets. Regions near the lateral ventricles all make strong contributions, making up 7 of the 15 highest ranking features. Substantial contributions can also be seen in frontal/association areas corresponding to the default mode, control, and salience/ventral attention networks. Areas near the hippocampus/fornix, thalamus, and somatomotor network also contribute. These findings are consistent when using the pretrained model for EDIS and SLABS (*Figure 5—figure supplement 1A and B*).

In contrast, with the children of GUSTO, notable differences can be found (*Figure 5C*). While the fornix is still the strongest contributor, it and other anterior areas near the ventricles (genu and body of corpus callosum, caudate) do not contribute as much. The overall prominence of white matter is also increased, especially in the brainstem (corticospinal tract and pontine crossing tract) and posterior regions (sagittal stratum, superior longitudinal fasciculus, and posterior limb of internal capsule). Furthermore, gray matter networks generally decrease in prominence, except the visual and limbic networks, which increase in prominence. The hippocampus, amygdala, and thalamus continue to make substantial contributions. Unlike in elderly, these features are not consistent when using the pretrained model (*Figure 5—figure supplement 1C*).

## Discussion

Our findings are the first, to our knowledge, to show the age-dependent generalizability of a pretrained brain age model to non-Caucasian participants, specifically Singaporean children and elderly. We also present novel results on the informativeness of longitudinal changes in brain age, independent of baseline brain age, to future executive function in healthy participants. Finally, we show that accurate models focus on distinct features in elderly and children, suggesting that the brain age model can extract relevant age-related information.

### Generalizability of pretrained brain age models to local datasets may be age-dependent

Overall, our results suggest the pretrained SFCN model could be directly applied to Singaporean elderly participants, but it needed to be finetuned for Singaporean children. Previous work with the SLABS dataset showed that aging-related changes in Chinese Singaporean elderly were comparable to previous studies conducted with primarily Caucasian participants (*Leong et al., 2017*). However, it was not initially clear whether this would carry over to a multidimensional index like brain age or to elderly datasets that better reflect the ethnic diversity of Singapore. Encouragingly, we found high accuracy in predicting age (i.e., low MAE and high correlation) in both EDIS (Chinese, Malay, and Indian participants) and SLABS (Chinese participants only) using the pretrained model. Furthermore, our results after finetuning were generally consistent with the original findings in elderly. This suggests that similarities were not specific to the SLABS sample, but could generalize to Singaporean elderly as a whole. In addition to similar aging patterns, the success of the pretrained model in this age range can be attributed to the abundance of training data around 60–80 years old, mostly from UKBiobank (*Leonardsen et al., 2022*).

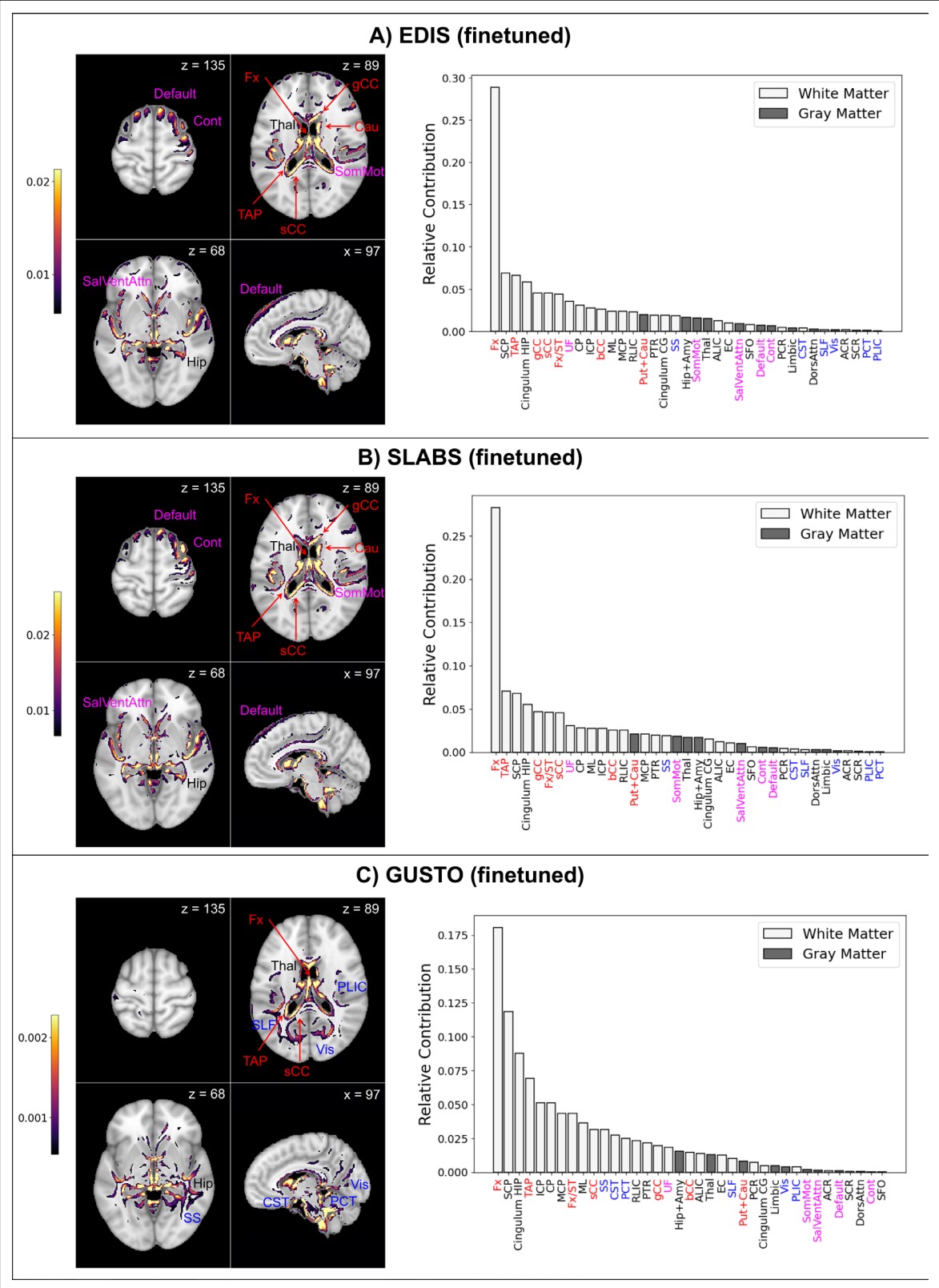

**Figure 5.** Finetuned brain age models focus on distinct features in children and elderly participants. The top 10% of features are shown for four representative brain slices on the left. Relative contributions for gray and white matter features across the whole brain are shown on the right. Regions near the lateral ventricles are labeled in red. Features more prominent in elderly than children are labeled in magenta, while features more prominent in children than elderly are labeled in blue. Features and relative contributions are generally consistent between (**A**) EDIS and (**B**) SLABS,

*Figure 5 continued*

but key differences can be seen in (**C**) GUSTO. EDIS, Epidemiology of Dementia in Singapore; SLABS, Singapore-Longitudinal Aging Brain Study; GUSTO,Growing Up in Singapore Towards healthy Outcomes; MCP,–middle cerebellar peduncle; PCT, Pontine crossing tract; gCC, genu of corpus callosum; bCC, body of corpus callosum; sCC, splenium of corpus callosum; Fx, fornix (column and body); CST, corticospinal tract; ML, medial lemniscus; ICP, inferior cerebellar peduncle; SCP, superior cerebellar peduncle; CP, cerebral peduncle; ALIC, anterior limb of internal capsule; PLIC, posterior limb of internal capsule; RLIC, retrolenticular part of internal capsule; ACR, anterior corona radiata; SCR, superior corona radiata; PCR, posterior corona radiata; PTR, posterior thalamic radiation; SS, sagittal stratum; EC, external capsule; cingulum CG, cingulum (cingulate gyrus); cingulum HIP, cingulum (hippocampus); Fx/ST, fornix (cres)/stria terminalis; SLF, superior longitudinal fasciculus; SFO, superior fronto-occipital fasciculus; UF, uncinate fasciculus; TAP, tapetum; Vis, visual network; SomMot, somatomotor network; DorsAttn, dorsal attention network; SalVentAttn, salience/ventral attention network; Limbic, limbic network; Cont , control/frontoparietal network; Default, default mode network; Hip+Amy, hippocampus + amygdala; Put+Cau, putamen + caudate; Tha l, thalamus.

The online version of this article includes the following figure supplement(s) for figure 5:

**Figure supplement 1.** Pretrained models focus on similar features as finetuned models in EDIS and SLABS, but not in GUSTO.

However, we found the pretrained model did not perform as well in children (i.e., low correlation). While the MAE was actually lower in children than elderly, this was likely due to the smaller age range (*de Lange et al., 2022*). After finetuning, the MAE reduced dramatically, further demonstrating the inadequacy of the pretrained model in this case. Previous work has indicated that brain structural differences between Chinese Singaporeans and non-Asian Americans may be more pronounced in young adults than elderly (*Chee et al., 2011*). This could conceivably extend to childhood and other ethnicities. However, another important factor is the model training age distribution, which only included 147 participants 5 years old or younger and had its earliest (and smallest) peak around 10 years old (*Leonardsen et al., 2022*). Notably, the pretrained model tended to predict all GUSTO ages around 10 years old, suggesting it may have been impacted by this imbalanced distribution.

Fortunately, finetuning the model produced distinct age groups, along with higher correlation and lower MAE. As discussed below, finetuning the model in children also shifted feature saliency and revealed a significant association with future executive function that was not found using the pretrained model. This suggests the model underwent a greater change in children, compared to elderly, to become both more accurate and meaningful. Furthermore, we found that the variance of finetuned predictions was the lowest at 4.5 years old and increased steadily with age, consistent with previous reports (*Brown et al., 2012*). This implies the 'brain maintenance' account of aging, where individuals start with the same or similar offsets, and different slopes result in increased variability over the lifespan (*Walhovd et al., 2023b*). This also suggests that the variance in brain age predictions at later ages is likely due to stable, lifelong factors, as well as ongoing changes. Thus, looking at longitudinal changes in brain age could help separate these influences.

## Longitudinal changes in brain age are informative of future executive function

Our results with baseline BAG were largely consistent with previous work in elderly. With a large sample of community-dwelling, non-demented participants from EDIS, we found significant associations with baseline cognition across multiple cognitive domains, consistent with a recent review (*Wrigglesworth et al., 2021*). Furthermore, with a smaller longitudinal sample of healthy participants from SLABS, we matched previous work finding an association with future decline in cognition, despite being not significant after multiple comparison correction (*Wrigglesworth et al., 2022*). In GUSTO children, we did not find a significant association between baseline BAG and baseline cognition at 4.5 years old or future cognition at 8.5 years old. To our knowledge, while brain age associations with cognition have been reported in samples spanning 3–22 years old (*Lewis et al., 2018*; *Jirsaraie et al., 2023b*; *Khundrakpam et al., 2015*; *Erus et al., 2015*), they have not been explored around the early age of 4.5 years old specifically. This is notable since our generalization analyses revealed that, after finetuning, brain age variability is lowest at this age and increases with age. Thus, cross-sectional associations with cognition may only occur at later ages, when there is more variability in brain age. Alternatively, more complex models may be needed to reveal cross-sectional structure–cognition association at such young ages.

Although we generally did not find associations between baseline BAG and future cognition in healthy participants, the story was different when including the early rate of change in BAG. Due to

the lack of available longitudinal cognitive data in GUSTO, we could only look at future cognition at a single time point instead of the rate of change. This likely introduced noise into our analyses as it did not account for the initial variation in cognition. The future cognitive scores in GUSTO were also all related to executive function, so we could not investigate other cognitive domains in children. Interestingly, within subdomains of executive function in children, we found the strongest association with inhibition, which has been proposed as a primary driver of domain-general executive function (*Tervo-Clemmens et al., 2023*). Thus, we found that early longitudinal changes in BAG associated with future executive function performance in both children and elderly, independent of baseline BAG. This is notable in light of recent evidence that early-life factors may affect cross-sectional brain measurements, but not longitudinal changes (*Walhovd et al., 2023a*). While the association was specific to executive function in elderly, it is presently not clear whether this was biased by the modest sample size of 6- to 10-year longitudinal data or whether brain age is particularly sensitive to this domain (*Wrigglesworth et al., 2021*). Previous work found associations between early- or mid-life factors and cross-sectional brain age (*Vidal-Pineiro et al., 2021*; *Elliott et al., 2021*; *Franke et al., 2018*), but associations with the longitudinal change of brain age were not investigated.

Our findings thus suggest that early-life factors could influence the cross-sectional brain age, but they are not the only influence throughout the lifespan. Early life factors may dominate the (relatively low) inter-individual variability in brain age at 4.5 years old. However, as normal development occurs in the next several years, children mature at different rates, leading to increasing variability in brain age predictions. This variance is also related to individual differences in executive function. In late life, these changes have accumulated to produce more variable brain age predictions. However, baseline brain age predictions do not associate with future cognitive performance, possibly since they reflect past factors. Information about ongoing changes in brain aging is needed to reveal associations with future rates of executive function decline. Taken together, these findings suggest that brain age, when measured longitudinally, can capture ongoing processes of healthy aging.

While early longitudinal changes in BAG associated with future executive function performance in both elderly and children, one notable difference lies in the direction of association. In elderly, increases in BAG were associated with worse executive function decline. In children, increases in BAG were associated with better future inhibitory performance. There have been somewhat conflicting reports on the direction of association between BAG and cognition in youth (*Lewis et al., 2018*; *Jirsaraie et al., 2023b*; *Khundrakpam et al., 2015*; *Erus et al., 2015*). However, our results are unique in examining the longitudinal change in BAG rather than the cross-sectional BAG. Thus, our results could reflect previously reported cognitive decline with increasing age in late life (*Hedden and Gabrieli, 2004*) and cognitive gains with increasing age in early life (*Roalf et al., 2014*). One of the few longitudinal studies in development related white matter and executive function development, and found that white matter growth in adolescence was associated with better inhibitory control, while growth in adulthood was associated with worse performance (*Simmonds et al., 2014*). Our brain age paradigm, based on multivariate features of the brain, further supports these findings in children and elderly. Specifically, a faster increase in BAG may imply that a child is developing ahead of schedule, resulting in more rapidly maturing cognitive functioning. Conversely, a slower increase in BAG at an older age may reflect mechanisms of brain maintenance at work, prolonging a more 'youthful' brain and sustained optimal cognitive performance.

## Salient features of the brain age model differ between elderly and children

Our work builds on recent efforts to interpret deep learning brain age models in aging (*Wood et al., 2022*; *Hofmann et al., 2022*; *Yin et al., 2023*). While the datasets, brain age models, and interpretability methods all differed among these studies, the most consistent finding was the importance of the lateral ventricles in elderly. This was evident in our models as well. Like other popular methods for extracting feature importance from deep learning models, our guided backpropagation method tended to highlight boundaries between regions and tissue types (i.e., edges). Thus, strong contributions from white matter areas such as the fornix and corpus callosum were likely at least partly due to the size of the lateral ventricles. These regions all ranked highly in elderly, suggesting the overall importance of the lateral ventricles.

Our other findings in elderly also broadly align with prior research in aging. We find important contributions around subcortical regions and frontal/association areas that are observed to degenerate more prominently in aging (*Fjell and Walhovd, 2010*). Among areas near the lateral ventricles, the fornix particularly stands out as the strongest contributor. This could be due to its connections with the hippocampus, suggesting fornix contributions may also reflect age-related hippocampus atrophy. Fornix was previously the strongest contributor in brain age models focused on white matter derived from diffusion MRI, and it showed the highest absolute correlations with age (*Korbmacher et al., 2023*). Corpus callosum and cerebellar peduncle were also found to strongly contribute in a separate white matter brain age model (*Mwangi et al., 2013*). Additionally, the importance of the thalamus, putamen and caudate, ventral attention network, and somatomotor network could indicate the importance of frontostriatal circuits. Frontostriatal changes have been proposed as a hallmark of healthy aging (*Fjell and Walhovd, 2010*), and the role of these and related regions (*Rodríguez-Nieto et al., 2022*) in supporting executive function could underlie the observed association between BAG and executive function.

We also show clear differences in feature importance between elderly and children, in line with prior research in development. The consistency between elderly datasets reinforces that these differences are not simply artifacts of using a different dataset. Most strikingly, we find evidence of a posterior to anterior pattern (*Marsh et al., 2008*; *Gogtay et al., 2004*) going from childhood to elderly. For instance, posterior areas near the lateral ventricles (tapetum and splenium of corpus callosum) continue to rank highly, while more anterior areas decrease in prominence. We also find a general increase in the relative importance of white matter, with greater increases in posterior regions. This is in line with a previous brain age model showing stronger contributions of white matter relative to gray matter in youth (*Brown et al., 2012*). The focus on the development of white matter could also underlie the observed association with executive function development (*Simmonds et al., 2014*; *Bagautdinova et al., 2023*; *Goddings et al., 2021*). Finally, we find increased contributions from the brain stem, which is consistent with its large volume changes in youth (*Cao et al., 2015*).

## Limitations

Our study is not without limitations. While we find encouraging signs that the model generalizes to Singaporean elderly, we cannot completely rule out more subtle issues that may have arisen from applying the model to these participants. For instance, finetuning the model slightly increased prediction accuracy and generally strengthened associations with cognition in elderly, suggesting the pretrained model may not have performed optimally. Furthermore, we have not tested the model in other non-Caucasian participants, which would be needed for a more comprehensive test of generalizability. The current study also only includes participants from very early and late life. Thus, future work would be needed to extend our results across the lifespan, with more participants and even longer follow-up times, in order to achieve a more complete picture.

## Conclusion

Here, we used a previously published brain age model to reveal age-dependent generalization to Asian participants, as well as age-dependent associations and interpretability of brain age. Specifically, we found the brain age model could be directly applied to Singaporean older adults, but it needed to be finetuned for Singaporean children. Furthermore, longitudinal changes in brain age were related to future executive function in both children and elderly participants. However, the direction of association was positive in children and negative in elderly. Combined with the identified salient features for brain age prediction, we conclude that increased brain age in early life could indicate more mature development, especially in white matter and posterior areas. Conversely, increased brain age in late life could suggest greater degeneration, especially around the lateral ventricles and frontal areas. Our results provide early evidence of the generalization capability of the brain age model and the ability of longitudinal measurements to capture ongoing aging process in the brain.

## Materials and methods
### Sample characteristics
We analyzed three datasets from Singapore, which are detailed in the following.

## Participants

### EDIS

EDIS was a cross-sectional study to measure the prevalence of cognitive impairment and dementia in Singapore, which has been described previously (*Hilal et al., 2013*; *Hilal et al., 2017*; *Wong et al., 2019*). We analyzed T1 MRI and cognitive data from 694 community-dwelling older adults. The same participants were used for all analyses. Ethics approval for the EDIS study was obtained from the Singapore Eye Research Institute and the National Healthcare Group Domain Specific Review Board. The study was conducted in accordance with the Declaration of Helsinki. Written informed consent was obtained, in the preferred language of participants, by bilingual study coordinators prior to recruitment into the study.

### SLABS

SLABS was a longitudinal, community-based study to characterize age-related brain changes and cognitive performance in healthy elderly in Singapore, which has been described previously (*Chee et al., 2009*). Participants underwent at most five phases of neuroimaging and neuropsychological assessments at approximately 2-year intervals. Neuropsychological assessments were performed within 3 months of neuroimaging. To test prediction accuracy of the brain age model, we first used 598 T1 scans from N=215 participants with MMSE score ≥26 at baseline (mean follow-up time = 4.0±3.3years). To investigate longitudinal associations in healthy elderly, we identified a subset of N=81 participants with (1) longitudinal T1 and cognitive data in the first three phases; (2) additional cognitive data in the last two phases (to study future cognitive decline, see *Figure 1C*); and (3) MMSE score ≥26 at baseline. Thus, mean follow-up time in this subset was 3.6±0.8 years for T1 scans (total number = 228), while mean follow-up time was 7.8± 1.0 years for cognitive scores (total number = 355). The study was approved by the Institutional Review Board of the National University of Singapore. All participants provided written informed consent prior to participation.

### GUSTO

GUSTO is a longitudinal birth cohort study to characterize early development in Singapore, which has been described previously (*Soh et al., 2014*). Participants were scanned at 4.5, 6.0, 7.5, and 10.5 years old. Neuropsychological assessments were taken at 4.5 and 8.5 years old. To test prediction accuracy of the brain age model in children, we first used 1702 T1 scans from N=678 normally developing children (mean follow-up time = 3.5±2.4 years). To investigate cross-sectional and longitudinal associations in healthy children, we identified subsets of participants (N=217–239) with the requisite imaging and cognitive data available, similar to SLABS. For the cross-sectional analysis, this included participants with both T1 and cognitive data at 4.5 years old (N=217). For the longitudinal analyses, this included participants with longitudinal T1 data from 4.5 to 7.5 years old and cognitive data at 8.5 years old (N=220 or 239). The study was approved by the National Healthcare Group Domain Specific Review Board (NHG DSRB) and the Sing Health Centralized Institutional Review Board (CIRB). Written informed consent was obtained from mothers. When children reached 6 years of age, children also provided oral consent.

## Neuropsychological assessments

### EDIS

Trained research psychologists administered a neuropsychological battery locally validated for Singaporean elderly, as described previously (*Hilal et al., 2013*). Briefly, the battery assessed the following seven cognitive domains using the corresponding tests: (1) Executive function: Frontal Assessment Battery and Maze Task; (2) Attention: Digit Span, Visual Memory Span, and Auditory Detection; (3) Language: Boston Naming Test and Verbal Fluency; (4) Visuomotor speed: Symbol Digit Modality Test and Digit Cancellation; (5) Visuoconstruction: Weschler Memory Scale-Revised Visual Reproduction Copy task, Clock Drawing, and Weschler Adult Intelligence Scale-Revised subtest of Block Design; (6) Verbal memory: Word List Recall and Story Recall; and (7) Visual memory: Picture Recall and Weschler Memory Scale-Revised Visual Reproduction.

For each individual test, raw scores were transformed to standardized z-scores using the mean and SD of that test (across all of EDIS, not just the imaging sample included here). Composite z-scores

for each domain were obtained by averaging all individual test z-scores within that domain. These domain-specific z-scores were then standardized using their own mean and SD. A global cognition z-score was calculated by averaging over all domain-specific z-scores and standardized using its own mean and SD. CIND was defined as impairment in at least one cognitive domain using education adjusted cut-off values of 1.5 SDs below the established normal means on individual tests. Failure in at least half of the tests in a domain constituted failure in that domain. Note that CIND was not a formal clinical diagnosis.

### SLABS

Trained researchers administered neuropsychological assessments within 3 months of the neuroimaging scan, as described previously (*Leong et al., 2017*). Briefly, the following five cognitive domains were assessed using the corresponding tests: (1) Executive function: Categorical Verbal Fluency Test and Design Fluency Test in the Delis-Kaplan Executive Function System (Unlike previous studies, Trail Making Test B was not included in this study due to missing data in later phases); (2) Attention: Digit Span Test and Spatial Span Test in Wechsler Memory Scale-Third Edition; (3) Processing speed: Symbol Digit Modalities Test, Symbol Search Task in the Wechsler Memory Scale-Third Edition, and Trail Making Test A; (4) Verbal Memory: Rey Auditory Verbal Learning Test; and (5) Visuospatial Memory: Visual Paired Associates Test. Composite T-scores (T-score = (z-score × 10)+50) were obtained for each domain and for global cognition following a similar procedure as EDIS.

### GUSTO

To maintain consistency with EDIS and SLABS, we selected standardized cognitive summary scores measured at 4.5 (baseline) and 8.5 (future) years old. These included the Kaufman Brief Intelligence Test Second Edition (KBIT-2) composite IQ standard score, the Wisconsin Card Sorting Test (WCST) total errors standard score, and the Developmental Neuropsychological Assessment Second Edition (NEPSY-II) scaled domain scores. The KBIT-2 was administered at 4.5 years and is a measure of abbreviated intelligence for children and adults aged 4 years to 90 years of age. The WCST is a lab-based measure of set-shifting/cognitive flexibility and was administered at age 8.5 years. The NEPSY-II was administered at 8.5 years and consisted of a word interference task requiring working memory recall (i.e. naming) and a Stroop task requiring predominantly inhibition in one condition and switching in another condition.

## Image acquisition and preprocessing

### EDIS

MRI scans were performed on a 3T Siemens Magnetom Tim Trio System (Siemens, Erlangen, Germany) at the Clinical Imaging Research Centre, National University of Singapore. High-resolution T1-weighted structural MRI was acquired using magnetization-prepared rapid gradient echo (MPRAGE) sequence (192 continuous sagittal slices, TR/TE/TI = 2300/1.9/900 ms, flip angle = 9°, FOV = 256 × 256 mm$^2$, matrix = 256 × 256, isotropic voxel size = 1.0 × 1.0×1.0 mm$^3$, bandwidth = 240 Hz/pixel).

### SLABS

For the first three phases, MRI scans were performed on a 3T Siemens Magnetom Tim Trio System (Siemens) at the Centre for Cognitive Neuroscience, Duke-NUS Medical School. High-resolution T1-weighted structural MRI was acquired using a MPRAGE sequence (192 continuous sagittal slices, TR/TE/TI = 2300/2.98/900 ms, flip angle = 9°, FOV = 256 × 240 mm$^2$, matrix = 256 × 240, isotropic voxel size = 1.0 × 1.0×1.0 mm$^3$, bandwidth = 240 Hz/pixel).

For the last two phases, following a scanner upgrade, MRI scans were performed on a 3T Siemens Magnetom Prisma Fit System (Siemens). High-resolution T1-weighted structural MRI was again acquired using a MPRAGE sequence (192 continuous sagittal slices, TR/TE/TI = 2300/2.28/900 ms, flip angle = 8°, FOV = 256 × 240 mm$^2$, matrix = 256 × 240, isotropic voxel size = 1.0 × 1.0×1.0 mm$^3$, bandwidth = 200 Hz/pixel).

### GUSTO

For scans taken at 4.5 and 6.0 years, MRI scans were performed on a 3T Siemens Magnetom Skyra System (Siemens) at KK Women's and Children's Hospital. High-resolution T1-weighted structural MRI

was acquired using a MPRAGE sequence (192 continuous sagittal slices, TR/TE/TI = 2000/2.08/877 ms, flip angle = 9°, FOV = 192 × 192 mm$^2$, matrix = 192 × 192, isotropic voxel size = 1.0 × 1.0×1.0 mm$^3$).

For scans taken at 7.5 and 10.5 years, MRI scans were performed on a 3T Siemens Magnetom Prisma Fit System (Siemens) at the National University of Singapore. The scanning parameters were the same as for 4.5 and 6.0 years.

## Preprocessing

For all datasets, we used the minimal preprocessing pipeline performed on the SFCN training set, as described previously (*Leonardsen et al., 2022*). Briefly, images were first skull-stripped with FreeSurfer (*Ségonne et al., 2004*), then reoriented to standard FMRIB (Oxford Centre for Functional MRI of the Brain) Software Library (FSL) (*Jenkinson et al., 2012*) orientation and linearly registered to Montreal Neurological Institute (MNI) 152 space using the FSL linear registration tool (FLIRT) (*Jenkinson and Smith, 2001*). Images were then cropped to 167 × 212 × 160 voxels, and voxel intensity values were normalized between 0 and 1. These minimally preprocessed images were input to the SFCN brain age model (*Figure 1A*). Similar to the original model (*Leonardsen et al., 2022*), we adopted a lenient manual quality control before conducting analyses, removing scans where a significant portion of the brain was missing or there was a registration failure. This excluded 2 scans/participants from EDIS and 12 scans from 11 participants from GUSTO.

## Brain age predictions

After preprocessing, we directly applied the pretrained brain age model (*Leonardsen et al., 2022*) to each of the datasets (all baseline and follow-up data) to generate brain age predictions. We used the regression variant of SFCN due to its superior generalization performance (*Leonardsen et al., 2022*). Performance was assessed using Pearson's correlation and mean absolute error (MAE) between brain age and chronological age. The model was considered to have performed well if both correlation was high and MAE was low. BAG was calculated by subtracting chronological age from brain age.

We then finetuned the model on each dataset separately to mitigate effects from the domain shift (i.e., the change in distribution from training data to testing data). We used all scans from each dataset without any exclusions. We split scans into 10 cross-validation folds, where each participant was included in the testing set exactly once. Of the remaining data for each fold, 80% was used for training and 20% was used for validation. In the case of longitudinal data, it was ensured that all scans from the same participant were kept in either the training, validation, or test set to avoid biased estimates. We used the pretrained model weights as initialization, then retrained all layers (*Figure 1B*).

We built on the original model code using the Keras (*Chollet, 2015*) interface of Tensorflow 2.11 (*Abadi et al., 2015*). We used the Adam optimizer with mean squared error loss. Upon recommendation of the original study authors, we set the dropout rate = 0.3 and weight decay = 1e-3. We selected the initial learning rate from {1e-3, 1e-4, 1e-5} using the validation sets of each fold. *Appendix 1—table 1* shows the optimal initial learning rate for each study and fold. We used a cosine learning rate decay over 25 epochs and trained the models for 35 epochs total. The final weights were taken from the epoch with the lowest validation MAE. Models were trained on a NVIDIA RTX 3090 GPU with 24 GB RAM on top of cuda 11.0 with a batch size of 4. More details on the finetuning process, including parameter adjustments and choices, can be found in Appendix 1.

## Associations with cognition

To examine cross-sectional and longitudinal associations with cognition in both elderly and children, we conducted several analyses, which are shown schematically in *Figure 1C*. *Appendix 2—table 2* shows the model equations. Statistical results were corrected for multiple comparisons across cognitive domains using the Holm-Bonferroni method (*Holm, 1979*). Change in adjusted $R^2$ ($\Delta R^2_{adj}$) was calculated from the difference between a model including the variable of interest and covariates and a model only including covariates. Variance inflation factors were confirmed to be less than five to rule out multicollinearity among baseline BAG, change in BAG (when included), and other covariates (especially chronological age). Analyses were performed in R 4.2.1 (*R Development Core Team, 2022*) with RStudio (*RStudio Team, 2022*).

### Elderly

For each cognitive domain in EDIS, we related baseline BAG to baseline cognitive score, with chronological age, sex, and years of education as covariates. For the longitudinal analyses in SLABS, we first calculated annual rates of change in BAG and cognition using linear regressions with time for each participant. For each cognitive domain in SLABS, we related baseline BAG to long-term rate of cognitive change (calculated from all five phases). Next, again for each cognitive domain, we related early rate of BAG change (calculated from the first three phases) to long-term rate of cognitive change (calculated from all five phases). If this relation was significant for a domain, we lastly related early rate of BAG change (calculated from the first three phases) to future rate of cognitive change (calculated from the last measurement of BAG onward). All models included chronological age, sex, and years of education as covariates. Models with rate of BAG change also included baseline BAG as a covariate.

### Children

For cross-sectional analyses, we related baseline BAG to baseline cognition, both at 4.5 years old. For longitudinal analyses, we again calculated annual rates of change in BAG using linear regressions with time for each participant. We then related baseline BAG to future cognition and early rate of change in BAG to future cognition. Here, early rate of BAG change was calculated from 4.5 to 7.5 years, while future cognition was measured at 8.5 years. Chronological age and sex were included as covariates in all models. Models with rate of BAG change also included baseline BAG as a covariate.

## Model interpretability

For each dataset, we investigated model interpretability using all scans from the same participants as the associations with cognition. Guided backpropagation (*Springenberg et al., 2015*) was used to compute individual saliency maps for both the pretrained and finetuned models. Guided backpropagation was previously shown to give similar results as occlusion for a brain age model, at a higher resolution and lower computational cost (*Wood et al., 2022*). For finetuned models, the fold where the participant was included in the test set was used. Maps were registered to a common space using Advanced Normalization Tools (ANTs) (*Avants et al., 2008*). Specifically, for each participant, input (minimally preprocessed) images were nonlinearly registered to MNI 152 space using the default parameters. This transformation was then applied to each participants' saliency maps.

To identify brain features that contributed the most to brain age predictions, we first averaged saliency maps over all participants in a study. We then retained the top 10% of voxels and calculated gray and white matter network/regional contributions. We used the 400-area Schaefer parcellation (*Schaefer et al., 2014*) assigned to seven functional networks (*Yeo et al., 2011*) for cortical gray matter. We averaged over all voxels in all parcels for each network. For subcortical gray matter, we used the automated anatomical labeling atlas 3 (AAL3) (*Rolls et al., 2020*) to identify regions containing the hippocampus and amygdala, the putamen and caudate, and the thalamus. For white matter, we used the ICBM-DTI-81 atlas (*Mori et al., 2008*) with 48 ROIs. For all regions, we averaged over all voxels in both hemispheres. Contributions were normalized to sum to 1, giving the relative contribution. To visualize saliency maps in 2D, we set the maximum value to the 99th percentile and overlaid select slices (x=97, z=68, z=89, z=135) over a MNI 152 template brain.

## Materials availability

Full 3D saliency maps generated from model interpretability are publicly available at https://figshare.com/articles/preprint/Brain_age_saliency_maps/24805545.

## Acknowledgements

We would like to thank all participants and the research teams of GUSTO, EDIS, and SLABS for their contributions. We would also like to thank Esten Leonardsen and team for making the SFCN pretrained model and code available and for providing suggestions on hyperparameters. Finally, we would like to thank Zijiao Chen, Yilei Wu, Yichi Zhang, Yao Feng Chong, and Joanna Su Xian Chong for helpful discussions. This study is supported by the Singapore National Medical Research Council (NMRC/OFLCG19May0035, NMRC/CIRG/1485/2018, NMRC/CSA-SI/0007/2016, NMRC/MOH-00707-01, NMRC/CG/435 M009/2017- NUH/NUHS, CIRG21nov-0007 and HLCA23Feb-0004), RIE2020 AME

Programmatic Fund from A*STAR, Singapore (No. A20G8b0102), Ministry of Education (MOE-T2EP40120-0007 &. T2EP2-0223-0025, MOET2EP20220-0001), and Yong Loo Lin School of Medicine Research Core Funding, National University of Singapore, Singapore. The Epidemiology of Dementia in Singapore (EDIS) study is supported by the National Medical Research Council (NMRC), Singapore (NMRC/CG/NUHS/2010 [Grant no: R-184-006-184-511]) and Bright Focus Foundation [R-608-000-248-597]. The research conducted in this study is also supported by the Singapore National Research Foundation under the Translational and Clinical Research (TCR) Flagship (GUSTO), Healthy Longevity Catalyst Awards (Zhou), Open-fund Young Individual Research Grant (Ng), Open Fund Large Collaborative Grant (OFLCG) Programmes (Zhou) by Singapore Ministry of Health's National Medical Research Council (NMRC) (Singapore – NMRC/TCR/004-NUS/2008; NMRC/TCR/012- NUHS/2014; HLCA23Feb0004; OFLCG/MOH-000504; MOH-OFYIRG19may-0012), National Medical Research Council Singapore Grants NMRC/STaR/0004/2008, NMRC/STaR/015/2013, and STAR19may-0001 (Chee), NMRC/CIRG/1446/2016 (Chen), NMRC/CIRG/1390/2014 and NMRC/CBRG/0088/2015 (Zhou), and from the Biomedical Research Council, Singapore (BMRC 04/1/36/372, Zhou). Additional funding is provided by the Duke-NUS Medical School Signature Research Program funded by Ministry of Health, Singapore, and Centre for Sleep and Cognition funded by Yong Loo Lin School of Medicine, National University of Singapore and the Brain–Body Initiative of A*STAR.

## Additional information

### Competing interests

Juan Helen Zhou: Reviewing editor, eLife. The other authors declare that no competing interests exist.

### Funding

| Funder | Grant reference number | Author |
|---|---|---|
| Singapore National Medical Research Council | NMRC/OFLCG19May-0035 | Juan Helen Zhou |
| Singapore National Medical Research Council | NMRC/CIRG/1485/2018 | Juan Helen Zhou |
| Singapore National Medical Research Council | NMRC/CSA-SI/0007/2016 | Juan Helen Zhou |
| Singapore National Medical Research Council | NMRC/MOH-00707-01 | Juan Helen Zhou |
| Singapore National Medical Research Council | NMRC/CG/435 M009/2017-NUH/NUHS | Juan Helen Zhou |
| Singapore National Medical Research Council | CIRG21nov-0007 | Juan Helen Zhou |
| Singapore National Medical Research Council | HLCA23Feb-0004 | Juan Helen Zhou |
| Singapore A*STAR | A20G8b0102 | Juan Helen Zhou |
| Singapore Ministry of Education | MOE-T2EP40120-0007 | Juan Helen Zhou |
| Singapore Ministry of Education | MOE-T2EP2-0223-0025 | Juan Helen Zhou |
| Singapore Ministry of Education | MOE-T2EP20220-0001 | Juan Helen Zhou |
| National University of Singapore | Yong Loo Lin School of Medicine Research Core Funding | Juan Helen Zhou |

The funders had no role in study design, data collection and interpretation, or the decision to submit the work for publication.

## Author contributions

Susan F Cheng, Conceptualization, Data curation, Software, Formal analysis, Visualization, Methodology, Writing – original draft, Writing – review and editing; Wan Lin Yue, Kwun Kei Ng, Xing Qian, Siwei Liu, Data curation, Writing – review and editing; Trevor WK Tan, Kim-Ngan Nguyen, Validation, Methodology; Ruth LF Leong, Saima Hilal, Ching-Yu Cheng, Ai Peng Tan, Evelyn C Law, Peter D Gluckman, Yap Seng Chong, Resources; Christopher Li-Hsian Chen, Resources, Funding acquisition, Writing – review and editing; Michael J Meaney, Resources, Funding acquisition; Michael WL Chee, Resources, Funding acquisition, Methodology, Writing – review and editing; BT Thomas Yeo, Juan Helen Zhou, Conceptualization, Resources, Supervision, Funding acquisition, Visualization, Methodology, Writing – review and editing

## Author ORCIDs

Susan F Cheng ⬚ https://orcid.org/0000-0002-1201-2880
Kwun Kei Ng ⬚ https://orcid.org/0000-0002-0584-7679
BT Thomas Yeo ⬚ https://orcid.org/0000-0002-0119-3276
Juan Helen Zhou ⬚ https://orcid.org/0000-0002-0180-8648

## Ethics

This work involved human subject data from three previously published studies. Ethics approval for the EDIS study was obtained from the Singapore Eye Research Institute and the National Healthcare Group Domain Specific Review Board. The study was conducted in accordance with the Declaration of Helsinki. Written informed consent was obtained, in the preferred language of participants, by bilingual study coordinators prior to recruitment into the study. The SLABS study was approved by the Institutional Review Board of the National University of Singapore. All participants provided written informed consent prior to participation. The GUSTO study was approved by the National Healthcare Group Domain Specific Review Board (NHG DSRB) and the Sing Health Centralized Institutional Review Board (CIRB). Written informed consent was obtained from mothers. When children reached 6 years of age, children also provided oral consent.

Joint Public Review: https://doi.org/10.7554/eLife.97036.3.sa1
Author response https://doi.org/10.7554/eLife.97036.3.sa2

---

# Additional files

## Supplementary files

MDAR checklist

## Data availability

Due to participant consent agreements, source data (deidentified or processed) that support the findings of this study are available from co-authors C.L.H.C (EDIS), M.W.L.C. (SLABS), and M.J.M. (GUSTO) upon collaborative request. The EDIS and SLABS data are available with a proper research agreement. Please direct inquiries to helen.zhou@nus.edu.sg. The GUSTO data is available on request, on approval by the GUSTO executive committee. Please direct inquiries to https://www.gusto.sg/request-to-collaborate/. Custom Python and R code for this study is available at https://github.com/hzlab/2024_Cheng_Longitudinal_Brain_Age (copy archived at *Cheng, 2025*). Some code was reused from https://github.com/estenhl/pyment-public/releases/tag/v1.0.0 (*Esten, 2021*).

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

# Appendix 1

## Finetuning process and details

Due to the high computational cost of model training, we employed an heuristic hyperparameter search. We started with GUSTO as it had the worst initial performance. We consulted the original model's author (Esten Leonardsen), who suggested tuning the last layer alone with dropout = 0.3, weight decay = 1e-3, and learning rate = 1e-3. We originally stopped model training once the validation MAE failed to improve after three epochs in a row. However, when tuning only the last layer, we found that the performance did not meaningfully improve and the learning curves showed evidence of underfitting (*Appendix 1—figure 1A*).

Thus, we next tried to tune all layers with the same hyperparameters. This resulted in much better performance, but the learning curves showed some evidence of instability (*Appendix 1—figure 1B*). Since Leonardsen previously observed instability with high learning rates, we tried to anneal the learning rate using cosine decay. Based on the shape of the previous training curve (*Appendix 1—figure 1B*), we set the number of decay steps to 25 epochs. This resulted in better performance and more stability (*Appendix 1—figure 1C*).

Having gotten satisfactory performance on GUSTO, we then tried applying the same hyperparameters to EDIS and SLABS. However, in these datasets, we observed that the model changed too quickly, leading to a very high validation error at the beginning of training and ultimately worse performance (*Appendix 1—figure 1D*). This was likely because the original pretrained model did not need to adjust as much to fit these datasets. Thus, we tried to reduce the initial learning rate by factors of 10, observing better training and performance at 1e-4 and 1e-5 (*Appendix 1—figure 1E*). We observed underfitting and worse performance at 1e-6 (*Appendix 1—figure 1F*). Thus, we optimized the initial learning rate out of {1e-3, 1e-4, 1e-5} for each fold in each dataset (*Appendix 1—table 1*). Observing that the model had well-converged by 35 epochs, we stopped training at 35 epochs. Since this resulted in good performance, we kept all other hyperparameters constant.

**Appendix 1—table 1.** Optimized initial learning rates for each dataset and fold.
EDIS, Epidemiology of Dementia in Singapore; SLABS, Singapore Longitudinal Aging Brain Study; GUSTO, Growing Up in Singapore Towards healthy Outcomes.

| Fold | EDIS | SLABS | GUSTO |
|---|---|---|---|
| 0 | 1e-4 | 1e-4 | 1e-3 |
| 1 | 1e-4 | 1e-4 | 1e-3 |
| 2 | 1e-4 | 1e-5 | 1e-3 |
| 3 | 1e-4 | 1e-5 | 1e-3 |
| 4 | 1e-5 | 1e-4 | 1e-3 |
| 5 | 1e-4 | 1e-4 | 1e-3 |
| 6 | 1e-5 | 1e-5 | 1e-3 |
| 7 | 1e-4 | 1e-5 | 1e-3 |
| 8 | 1e-4 | 1e-5 | 1e-3 |
| 9 | 1e-4 | 1e-4 | 1e-3 |

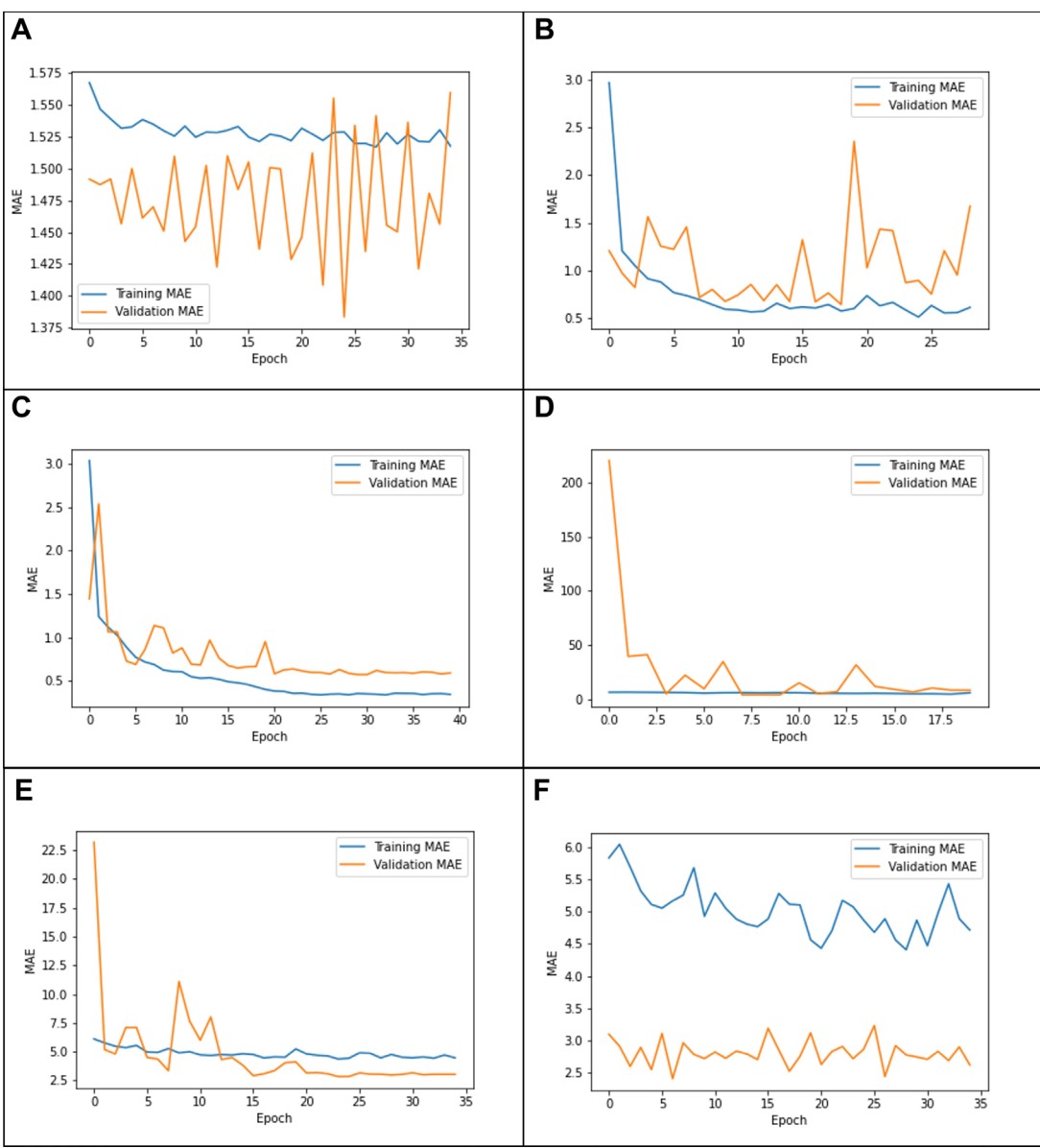

**Appendix 1—figure 1.** Example learning curves from (**A**) tuning the last layer only on Growing Up in Singapore Towards healthy Outcomes (GUSTO), showing underfitting; (**B**) tuning all layers on GUSTO, showing in stability; (**C**) using a cosine learning rate decay, showing a good fit (**D**) using the same parameters on Epidemiology of Dementia in Singapore (EDIS),showing 'forgetting'; (**E**) using a lower initial learning rate(1e-4) on EDIS, showing a better fit; and (**F**) using an initial learning rate of 1e-6, showing underfitting.

# Appendix 2

## Cognitive characteristics and model equations

**Appendix 2—table 1.** Participant cognitive characteristics at baseline.
EDIS was cross-sectional, while SLABS and GUSTO were longitudinal. EDIS, Epidemiology of Dementia in Singapore; SLABS, Singapore Longitudinal Aging Brain Study; GUSTO, Growing Up in Singapore Towards healthy Outcomes; KBIT-2, Kaufman Brief Intelligence Test Second Edition; WCST, Wisconsin Card Sorting Test; NEPSY-II, A Developmental Neuropsychological Assessment Second Edition.

| Characteristic | EDIS (N=694) |
|---|---|
| Global cognition z-score | $-2.55 \pm 2.21$ $(-10.47 - 1.96)$ |
| Executive function domain z-score | $-1.57 \pm 1.85$ $(-7.79 - 1.26)$ |
| Attention domain z-score | $-2.13 \pm 2.63$ $(-13.69 - 2.65)$ |
| Language domain z-score | $-1.84 \pm 1.70$ $(-11.72 - 2.85)$ |
| Visuomotor speed domain z-score | $-1.64 \pm 1.76$ $(-5.87 - 2.25)$ |
| Visuoconstruction domain z-score | $-2.50 \pm 2.57$ $(-13.63 - 2.37)$ |
| Verbal memory domain z-score | $-1.40 \pm 1.40$ $(-5.17 - 2.78)$ |
| Visual memory domain z-score | $-1.50 \pm 1.48$ $(-9.10 - 2.15)$ |

| Characteristic | SLABS (N=81) |
|---|---|
| Global cognition T-score | $51.82 \pm 4.75$ $(41.21 - 61.65)$ |
| Executive function domain T-score | $51.98 \pm 5.57$ $(42.39 - 64.08)$ |
| Attention domain T-score | $50.52 \pm 6.11$ $(39.62 - 65.04)$ |
| Processing speed domain T-score | $52.96 \pm 7.22$ $(36.13 - 71.84)$ |
| Verbal memory domain T-score | $52.18 \pm 7.70$ $(32.38 - 67.04)$ |
| Visuospatial memory domain T-score | $51.48 \pm 7.62$ $(35.00 - 65.80)$ |
| Cognition follow-up (years) | $7.83 \pm 0.97$ $(5.58 - 9.59)$ |

| Characteristic | GUSTO (N=217–239) |
|---|---|
| KBIT-2 Composite IQ Standard Score (N=217, 4.5 years old) | $92.38 \pm 14.24$ $(52 - 132)$ |
| WCST Total Errors Standard Score (N=220, 8.5 years old) | $99.45 \pm 15.95$ $(64 - 136)$ |
| NEPSY-II Naming Scaled Score (N=239, 8.5 years old) | $10.45 \pm 3.66$ $(3 - 19)$ |
| NEPSY-II Inhibition Scaled Score (N=239, 8.5 years old) | $10.31 \pm 3.3$ $(2 - 18)$ |
| NEPSY-II Switching Scaled Score (N=239, 8.5 years old) | $9.26 \pm 4.03$ $(1 - 19)$ |

**Appendix 2—table 2.** Model equations for analyzing associations with cognition.
Cog, standardized cognitive score; BAG, brain age gap; bl, baseline; $\Delta$, annual rate of change; EDIS, Epidemiology of Dementia in Singapore; SLABS, Singapore Longitudinal Aging Brain Study; GUSTO, Growing Up in Singapore Towards healthy Outcomes.

| Age | Dataset | Label | Equation |
|---|---|---|---|
| Elderly | EDIS | I. Baseline vs. baseline | $\mathrm{Cog}_{bl} = \mathrm{BAG}_{bl} + \mathrm{age} + \mathrm{sex} + \mathrm{education}$ |
| Elderly | SLABS | II. Baseline vs. change | $\Delta\mathrm{Cog}_{long} = \mathrm{BAG}_{bl} + \mathrm{age} + \mathrm{sex} + \mathrm{education}$ |
| Elderly | SLABS | III. Change vs. change (overlapping) | $\Delta\mathrm{Cog}_{long} = \Delta\mathrm{BAG}_{early} + \mathrm{BAG}_{bl} + \mathrm{age} + \mathrm{sex} + \mathrm{education}$ |

*Appendix 2—table 2 Continued on next page*

*Appendix 2—table 2 Continued*

| Age | Dataset | Label | Equation |
|-----|---------|-------|----------|
| Elderly | SLABS | IV. Change vs. change (future) | $\Delta\text{Cog}_{future} = \Delta\text{BAG}_{early} + \text{BAG}_{bl} + \text{age} + \text{sex} + \text{education}$ |
| Children | GUSTO | I. Baseline vs. baseline | $\text{Cog}_{4.5} = \text{BAG}_{4.5} + \text{age} + \text{sex}$ |
| Children | GUSTO | II. Baseline vs. future | $\text{Cog}_{8.5} = \text{BAG}_{4.5} + \text{age} + \text{sex}$ |
| Children | GUSTO | III. Change vs. future | $\text{Cog}_{8.5} = \Delta\text{BAG}_{4.5-7.5} + \text{BAG}_{4.5} + \text{age} + \text{sex}$ |

# Appendix 3

## Cognitive association results

**Appendix 3—table 1.** Pretrained baseline vs. baseline results from EDIS.
p-values are bolded if less than α = 0.05. BAG, brain age gap; EDIS, Epidemiology of Dementia in Singapore; β, standardized regression coefficient; $CI_L$, lower limit of 95% CI; $CI_U$, upper limit of 95% CI; p, uncorrected p-value; $p_{corr}$, corrected p-value; $\Delta R^2_{adj}$, change in adjusted $R^2$ when adding variable of interest; $R^2$, model coefficient of determination.

| Variable of interest | Outcome | β (CI$_L$, CI$_U$) | p | p$_{corr}$ | Δ $R^2_{adj}$ | $R^2$ |
|---|---|---|---|---|---|---|
| Baseline BAG (pretrained) | Baseline global cognition | –0.1125 (–0.17, –0.06) | <0.0001 | 0.0006 | 0.0106 | 0.5011 |
| Baseline BAG (pretrained) | Baseline executive function | –0.1029 (–0.17, –0.04) | 0.0019 | 0.0076 | 0.0085 | 0.3297 |
| Baseline BAG (pretrained) | Baseline attention | –0.0404 (–0.11, 0.03) | 0.2461 | 0.2461 | 0.0004 | 0.2562 |
| Baseline BAG (pretrained) | Baseline language | –0.1145 (–0.18, –0.05) | 0.0009 | 0.0047 | 0.0107 | 0.2677 |
| Baseline BAG (pretrained) | Baseline visuomotor speed | –0.0825 (–0.14, –0.02) | 0.0052 | 0.0136 | 0.0053 | 0.4684 |
| Baseline BAG (pretrained) | Baseline visuo-construction | –0.0896 (–0.15, –0.03) | 0.0045 | 0.0136 | 0.0063 | 0.3911 |
| Baseline BAG (pretrained) | Baseline verbal memory | –0.1096 (–0.17, –0.05) | 0.0006 | 0.0034 | 0.0099 | 0.3826 |
| Baseline BAG (pretrained) | Baseline visual memory | –0.1395 (–0.20, –0.08) | < 0.0001 | 0.0002 | 0.0165 | 0.3515 |

**Appendix 3—table 2.** Finetuned baseline vs. baseline results from EDIS.
p-values are bolded if less than α = 0.05. BAG, brain age gap; EDIS, Epidemiology of Dementia in Singapore; β, standardized regression coefficient; CI$_L$, lower limit of 95% CI; CI$_U$, upper limit of 95% CI; p, uncorrected p-value; $p_{corr}$, corrected p-value; $\Delta R^2_{adj}$, change in adjusted $R^2$ when adding variable of interest; $R^2$, model coefficient of determination.

| Variable of interest | Outcome | β (CI$_L$, CI$_U$) | p | p$_{corr}$ | Δ $R^2_{adj}$ | $R^2$ |
|---|---|---|---|---|---|---|
| Baseline BAG (finetuned) | Baseline global cognition | –0.1661 (–0.23, –0.10) | < 0.0001 | < 0.0001 | 0.0171 | 0.5076 |
| Baseline BAG (finetuned) | Baseline executive function | –0.1607 (–0.24, –0.08) | <0.0001 | 0.0002 | 0.0158 | 0.3369 |
| Baseline BAG (finetuned) | Baseline attention | –0.0632 (–0.14, 0.02) | 0.1223 | 0.1223 | 0.0015 | 0.2573 |
| Baseline BAG (finetuned) | Baseline language | –0.1535 (–0.23, –0.07) | 0.0002 | 0.0007 | 0.0142 | 0.2712 |
| Baseline BAG (finetuned) | Baseline visuomotor speed | –0.1071 (–0.17, –0.04) | 0.0020 | 0.0040 | 0.0066 | 0.4697 |
| Baseline BAG (finetuned) | Baseline visuoconstruction | –0.1291 (–0.20, –0.06) | 0.0005 | 0.0015 | 0.0099 | 0.3946 |
| Baseline BAG (finetuned) | Baseline verbal memory | –0.1722 (–0.24, –0.10) | <0.0001 | <0.0001 | 0.0183 | 0.3910 |
| Baseline BAG (finetuned) | Baseline visual memory | –0.2194 (–0.29, –0.15) | <0.0001 | <0.0001 | 0.0302 | 0.3651 |

**Appendix 3—table 3.** Pretrained baseline vs. baseline results from SLABS.
p-values are bolded if less than α = 0.05. BAG, brain age gap; SLABS, Singapore Longitudinal Aging Brain Study; β, standardized regression coefficient; CI$_L$, lower limit of 95% CI; CI$_U$, upper limit of 95% CI; p, uncorrected p-value; $p_{corr}$, corrected p-value; $\Delta R^2_{adj}$, change in adjusted $R^2$ when adding variable of interest; $R^2$, model coefficient of determination.

| Variable of interest | Outcome | β (CI$_L$, CI$_U$) | p | p$_{corr}$ | Δ$R^2_{adj}$ | $R^2$ |
|---|---|---|---|---|---|---|
| Baseline BAG (pretrained) | Baseline global cognition | –0.0574 (–0.18, 0.06) | 0.3376 | 1.0000 | –0.0002 | 0.3514 |
| Baseline BAG (pretrained) | Baseline executive function | –0.0447 (–0.16, 0.07) | 0.4600 | 1.0000 | –0.0015 | 0.3379 |
| Baseline BAG (pretrained) | Baseline verbal memory | –0.0089 (–0.14, 0.12) | 0.8925 | 1.0000 | –0.0038 | 0.2090 |
| Baseline BAG (pretrained) | Baseline visual memory | –0.1299 (–0.27, 0.01) | 0.0716 | 0.4297 | 0.0105 | 0.0643 |
| Baseline BAG (pretrained) | Baseline attention | –0.0187 (–0.15, 0.11) | 0.7750 | 1.0000 | –0.0035 | 0.2261 |
| Baseline BAG (pretrained) | Baseline processing speed | –0.0036 (–0.12, 0.12) | 0.9492 | 1.0000 | –0.0029 | 0.4038 |

**Appendix 3—table 4.** Finetuned baseline vs. baseline results from SLABS.
p-values are bolded if less than α = 0.05. BAG, brain age gap; SLABS, Singapore Longitudinal Aging Brain Study; β, standardized regression coefficient; CI$_L$, lower limit of 95% CI; CI$_U$, upper limit of 95% CI; p, uncorrected p-value; p$_{corr}$, corrected p-value; Δ$R^2_{adj}$, change in adjusted $R^2$ when adding variable of interest; $R^2$, model coefficient of determination.

| Variable of interest | Outcome | β (CI$_L$, CI$_U$) | p | p$_{corr}$ | Δ$R^2_{adj}$ | $R^2$ |
|---|---|---|---|---|---|---|
| Baseline BAG (finetuned) | Baseline global cognition | –0.0674 (–0.19, 0.05) | 0.2784 | 1.0000 | 0.0006 | 0.3522 |
| Baseline BAG (finetuned) | Baseline executive function | –0.0378 (–0.16, 0.09) | 0.5477 | 1.0000 | –0.0021 | 0.3373 |
| Baseline BAG (finetuned) | Baseline verbal memory | –0.0548 (–0.19, 0.08) | 0.4245 | 1.0000 | –0.0014 | 0.2113 |
| Baseline BAG (finetuned) | Baseline visual memory | –0.1243 (–0.27, 0.02) | 0.0974 | 0.5846 | 0.0081 | 0.0621 |
| Baseline BAG (finetuned) | Baseline attention | 0.0056 (–0.13, 0.14) | 0.9342 | 1.0000 | –0.0038 | 0.2258 |
| Baseline BAG (finetuned) | Baseline processing speed | 0.0229 (–0.14, 0.09) | 0.7008 | 1.0000 | –0.0025 | 0.4042 |

**Appendix 3—table 5.** Pretrained baseline vs. change results from SLABS.
p-values are bolded if less than α = 0.05. BAG, brain age gap; SLABS, Singapore Longitudinal Aging Brain Study; β, standardized regression coefficient; CI$_L$, lower limit of 95% CI; CI$_U$, upper limit of 95% CI; p, uncorrected p-value; p$_{corr}$, corrected p-value; Δ$R^2_{adj}$, change in adjusted $R^2$ when adding variable of interest; $R^2$, model coefficient of determination.

| Variable of interest | Outcome | β (CI$_L$, CI$_U$) | p | p$_{corr}$ | Δ$R^2_{adj}$ | $R^2$ |
|---|---|---|---|---|---|---|
| Baseline BAG (pretrained) | Change in global cognition | –0.1213 (–0.36, 0.12) | 0.3232 | 1.0000 | –0.0002 | 0.0211 |
| Baseline BAG (pretrained) | Change in executive function | –0.2477 (–0.48, –0.01) | **0.0406** | 0.2433 | 0.0424 | 0.0711 |
| Baseline BAG (pretrained) | Change in verbal memory | –0.1263 (–0.37, 0.11) | 0.2970 | 1.0000 | 0.0013 | 0.0490 |
| Baseline BAG (pretrained) | Change in visual memory | 0.0337 (–0.21, 0.28) | 0.7815 | 1.0000 | –0.0122 | 0.0338 |
| Baseline BAG (pretrained) | Change in attention | 0.0253 (–0.21, 0.26) | 0.8345 | 1.0000 | –0.0125 | 0.0457 |
| Baseline BAG (pretrained) | Change in processing speed | –0.1032 (–0.35, 0.14) | 0.4015 | 1.0000 | –0.0039 | 0.0157 |

**Appendix 3—table 6.** Finetuned baseline vs. change results from SLABS.
p-values are bolded if less than α = 0.05. BAG, brain age gap; SLABS, Singapore Longitudinal Aging Brain Study; β, standardized regression coefficient; CI$_L$, lower limit of 95% CI; CI$_U$, upper limit of 95% CI; p, uncorrected p-value; p$_{corr}$, corrected p-value; Δ$R^2_{adj}$, change in adjusted $R^2$ when adding variable of interest; $R^2$, model coefficient of determination.

| Variable of interest | Outcome | β (CI$_L$, CI$_U$) | p | p$_{corr}$ | Δ$R^2_{adj}$ | $R^2$ |
|---|---|---|---|---|---|---|
| Baseline BAG (finetuned) | Change in global cognition | –0.0480 (–0.30, 0.20) | 0.7041 | 1.0000 | –0.0116 | 0.0103 |

*Appendix 3—table 6 Continued on next page*

*Appendix 3—table 6 Continued*

| Variable of interest | Outcome | β (CI$_L$, CI$_U$) | p | p$_{corr}$ | Δ$R^2_{adj}$ | $R^2$ |
|---|---|---|---|---|---|---|
| Baseline BAG (finetuned) | Change in executive function | –0.1165 (–0.36, 0.13) | 0.3531 | 1.0000 | –0.0017 | 0.0292 |
| Baseline BAG (finetuned) | Change in verbal memory | –0.1431 (–0.39, 0.10) | 0.2491 | 1.0000 | 0.0045 | 0.0520 |
| Baseline BAG (finetuned) | Change in visual memory | 0.0409 (–0.21, 0.29) | 0.7431 | 1.0000 | –0.0118 | 0.0342 |
| Baseline BAG (finetuned) | Change in attention | 0.1232 (–0.13, 0.37) | 0.3193 | 1.0000 | 0.0001 | 0.0576 |
| Baseline BAG (finetuned) | Change in processing speed | –0.0125 (–0.26, 0.23) | 0.9214 | 1.0000 | –0.0134 | 0.0066 |

**Appendix 3—table 7.** Pretrained change vs. change results from SLABS.
p-values are bolded if less than α = 0.05. BAG, brain age gap; SLABS, Singapore Longitudinal Aging Brain Study; β, standardized regression coefficient; CI$_L$, lower limit of 95% CI; CI$_U$, upper limit of 95% CI; p, uncorrected p-value; p$_{corr}$, corrected p-value; Δ$R^2_{adj}$, change in adjusted $R^2$ when adding variable of interest; $R^2$, model coefficient of determination.

| Variable of interest | Outcome | β (CI$_L$, CI$_U$) | p | p$_{corr}$ | Δ$R^2_{adj}$ | $R^2$ |
|---|---|---|---|---|---|---|
| Change in BAG (pretrained) | Change in global cognition | –0.1415 (–0.39, 0.11) | 0.2689 | 1.0000 | 0.0033 | 0.0370 |
| Change in BAG (pretrained) | Change in executive function | –0.3807 (–0.61, –0.15) | **0.0017** | **0.0100** | 0.1100 | 0.1864 |
| Change in BAG (pretrained) | Change in verbal memory | –0.0312 (–0.28, 0.22) | 0.8056 | 1.0000 | –0.0125 | 0.0498 |
| Change in BAG (pretrained) | Change in visual memory | 0.0917 (–0.16, 0.34) | 0.4718 | 1.0000 | –0.0064 | 0.0405 |
| Change in BAG (pretrained) | Change in attention | –0.1872 (–0.44, 0.06) | 0.1371 | 0.6857 | 0.0164 | 0.0736 |
| Change in BAG (pretrained) | Change in processing speed | –0.0868 (–0.34, 0.17) | 0.5001 | 1.0000 | –0.0074 | 0.0217 |

**Appendix 3—table 8.** Finetuned change vs. change results from SLABS.
p-values are bolded if less than α = 0.05. BAG, brain age gap; SLABS, Singapore Longitudinal Aging Brain Study; β, standardized regression coefficient; CI$_L$, lower limit of 95% CI; CI$_U$, upper limit of 95% CI; p, uncorrected p-value; p$_{corr}$, corrected pvalue; Δ$R^2_{adj}$, change in adjusted $R^2$ when adding variable of interest; $R^2$, model coefficient of determination.

| Variable of interest | Outcome | β (CI$_L$, CI$_U$) | p | p$_{corr}$ | Δ$R^2_{adj}$ | $R^2$ |
|---|---|---|---|---|---|---|
| Change in BAG (finetuned) | Change in global cognition | –0.2367 (–0.48, 0.01) | 0.0576 | 0.2305 | 0.0360 | 0.0570 |
| Change in BAG (finetuned) | Change in executive function | –0.3861 (–0.62, –0.15) | **0.0014** | **0.0084** | 0.1190 | 0.1536 |
| Change in BAG (finetuned) | Change in verbal memory | –0.1017 (–0.35, 0.14) | 0.4091 | 1.0000 | –0.0041 | 0.0607 |
| Change in BAG (finetuned) | Change in visual memory | –0.0284 (–0.28, 0.22) | 0.8200 | 1.0000 | –0.0128 | 0.0349 |
| Change in BAG (finetuned) | Change in attention | –0.2651 (–0.50, –0.03) | **0.0287** | 0.1434 | 0.0493 | 0.1163 |
| Change in BAG (finetuned) | Change in processing speed | –0.0842 (–0.33, 0.17) | 0.5050 | 1.0000 | –0.0076 | 0.0125 |

**Appendix 3—table 9.** Pretrained baseline vs. future results from GUSTO.
p-values are bolded if less than α = 0.05. BAG, brain age gap; GUSTO, Growing Up in Singapore Towards healthy Outcomes; NEPSY-II, A Developmental Neuropsychological Assessment Second Edition; WCST, Wisconsin Card Sorting Test; β, standardized regression coefficient; CI$_L$, lower limit of

95% CI; CI$_U$, upper limit of 95% CI; p, uncorrected p-value; p$_{corr}$, corrected pvalue; $\Delta R^2_{adj}$, change in adjusted $R^2$ when adding variable of interest; $R^2$, model coefficient of determination.

| Variable of interest | Outcome | β (CI$_L$, CI$_U$) | p | p$_{corr}$ | $\Delta R^2_{adj}$ | $R^2$ |
|---|---|---|---|---|---|---|
| Baseline BAG (pretrained) | Future WCST Standard Score | –0.0585 (–0.24, 0.13) | 0.5334 | 0.9798 | –0.0028 | 0.0256 |
| Baseline BAG (pretrained) | Future Naming (NEPSY-II) | 0.1290 (–0.05, 0.31) | 0.1573 | 0.6292 | 0.0043 | 0.0149 |
| Baseline BAG (pretrained) | Future Inhibition (NEPSY-II) | 0.1074 (–0.07, 0.29) | 0.2387 | 0.7161 | 0.0017 | 0.0155 |
| Baseline BAG (pretrained) | Future Switching (NEPSY-II) | 0.0630 (–0.12, 0.24) | 0.4899 | 0.9798 | –0.0022 | 0.0106 |

**Appendix 3—table 10.** Finetuned baseline vs. future results from GUSTO.
p-values are bolded if less than α = 0.05. BAG, brain age gap; GUSTO, Growing Up in Singapore Towards healthy Outcomes; NEPSY-II, A Developmental Neuropsychological Assessment Second Edition; WCST, Wisconsin Card Sorting Test; β, standardized regression coefficient; CI$_L$, lower limit of 95% CI; CI$_U$, upper limit of 95% CI; p, uncorrected p-value; p$_{corr}$, corrected p-value; $\Delta R^2_{adj}$, change in adjusted $R^2$ when adding variable of interest; $R^2$, model coefficient of determination.

| Variable of interest | Outcome | β (CI$_L$, CI$_U$) | p | p$_{corr}$ | $\Delta R^2_{adj}$ | $R^2$ |
|---|---|---|---|---|---|---|
| Baseline BAG (finetuned) | Future WCST Standard Score | –0.0046 (–0.20, 0.20) | 0.9642 | 1.0000 | –0.0046 | 0.0239 |
| Baseline BAG (finetuned) | Future Naming (NEPSY-II) | –0.0108 (–0.21, 0.19) | 0.9143 | 1.0000 | –0.0042 | 0.0065 |
| Baseline BAG (finetuned) | Future Inhibition (NEPSY-II) | 0.0829 (–0.11, 0.28) | 0.4086 | 1.0000 | –0.0013 | 0.0125 |
| Baseline BAG (finetuned) | Future Switching (NEPSY-II) | –0.0359 (–0.23, 0.16) | 0.7203 | 1.0000 | –0.0037 | 0.0091 |

**Appendix 3—table 11.** Pretrained change vs. future results from GUSTO.
p-values are bolded if less than α = 0.05. BAG, brain age gap; GUSTO, Growing Up in Singapore Towards healthy Outcomes; NEPSY-II, A Developmental Neuropsychological Assessment Second Edition; WCST, Wisconsin Card Sorting Test; β, standardized regression coefficient; CI$_L$, lower limit of 95% CI; CI$_U$, upper limit of 95% CI; p, uncorrected p-value; p$_{corr}$, corrected p-value; $\Delta R^2_{adj}$, change in adjusted $R^2$ when adding variable of interest; $R^2$, model coefficient of determination.

| Variable of interest | Outcome | β (CI$_L$, CI$_U$) | p | p$_{corr}$ | $\Delta R^2_{adj}$ | $R^2$ |
|---|---|---|---|---|---|---|
| Change in BAG (pretrained) | Future WCST Standard Score | –0.0639 (–0.21, 0.08) | 0.3877 | 1.0000 | –0.0011 | 0.0290 |
| Change in BAG (pretrained) | Future Naming (NEPSY-II) | –0.0280 (–0.17, 0.12) | 0.7020 | 1.0000 | –0.0036 | 0.0156 |
| Change in BAG (pretrained) | Future Inhibition (NEPSY-II) | 0.0376 (–0.11, 0.18) | 0.6073 | 1.0000 | –0.0031 | 0.0166 |
| Change in BAG (pretrained) | Future Switching (NEPSY-II) | 0.0021 (–0.14, 0.15) | 0.9772 | 1.0000 | –0.0043 | 0.0106 |

**Appendix 3—table 12.** Finetuned change vs. future results from GUSTO.
p-values are bolded if less than α = 0.05. BAG, brain age gap; GUSTO, Growing Up in Singapore Towards healthy Outcomes; NEPSY-II, A Developmental Neuropsychological Assessment Second Edition; WCST, Wisconsin Card Sorting Test; β, standardized regression coefficient; CI$_L$, lower limit of 95% CI; CI$_U$, upper limit of 95% CI; p, uncorrected p-value; p$_{corr}$, corrected pvalue; $\Delta R^2_{adj}$, change in adjusted $R^2$ when adding variable of interest; $R^2$, model coefficient of determination.

| Variable of interest | Outcome | β (CI$_L$, CI$_U$) | p | p$_{corr}$ | $\Delta R^2_{adj}$ | $R^2$ |
|---|---|---|---|---|---|---|
| Change in BAG (pretrained) | Future WCST Standard Score | –0.0091 (–0.16, 0.14) | 0.9052 | 1.0000 | –0.0045 | 0.0239 |
| Change in BAG (pretrained) | Future Naming (NEPSY-II) | 0.0158 (–0.14, 0.17) | 0.8413 | 1.0000 | –0.0041 | 0.0067 |
| Change in BAG (pretrained) | Future Inhibition (NEPSY-II) | 0.2006 (0.05, 0.35) | **0.0103** | **0.0411** | 0.0237 | 0.0400 |
| Change in BAG (pretrained) | Future Switching (NEPSY-II) | 0.1795 (0.03, 0.33) | **0.0221** | 0.0663 | 0.0181 | 0.0311 |

