## [Editor Report · eLife Assessment]

This **valuable** study marks a significant advancement in brain aging research by centering on Asian populations (Chinese, Malay, and Indian Singaporeans), a group frequently underrepresented in such studies. It unveils **solid** evidence for anatomical differences in brain aging predictors between the young and old age groups. Overall, this study broadens our understanding of brain aging across diverse ethnicities.

---

## [Referee Report · Joint Public Review]

Summary:

The authors of the study investigated the generalization capabilities of a deep learning brain age model across different age groups within the Singaporean population, encompassing both elderly individuals aged 55 to 88 years and children aged 4 to 11 years. The model, originally trained on a dataset primarily consisting of Caucasian adults, demonstrated a varying degree of adaptability across these age groups. For the elderly, the authors observed that the model could be applied with minimal modifications, whereas for children, significant fine-tuning was necessary to achieve accurate predictions. Through their analysis, the authors established a correlation between changes in the brain age gap and future executive function performance across both demographics. Additionally, they identified distinct neuroanatomical predictors for brain age in each group: lateral ventricles and frontal areas were key in elderly participants, while white matter and posterior brain regions played a crucial role in children. These findings underscore the authors' conclusion that brain age models hold the potential for generalization across diverse populations, further emphasizing the significance of brain age progression as an indicator of cognitive development and aging processes.

Strengths:

(1) The study tackles a crucial research gap by exploring the adaptability of a brain age model across Asian demographics (Chinese, Malay, and Indian Singaporeans), enriching our knowledge of brain aging beyond Western populations.

(2) It uncovers distinct anatomical predictors of brain aging between elderly and younger individuals, highlighting a significant finding in the understanding of age-related changes and ethnic differences.

In summary, this paper underscores the critical need to include diverse ethnicities in model testing and estimation.

Comments on revisions:

The previously mentioned weaknesses were addressed in the revision process. As stated earlier the paper tackles a crucial research gap by exploring the adaptability of a brain-age model across Asian demographics (Chinese, Malay, and Indian Singaporeans), enriching our knowledge of brain aging beyond Western populations.

---

## [Author Response]

The following is the authors’ response to the original reviews.

**Reviewer #1 (Public Review):**
The authors of the study investigated the generalization capabilities of a deep learning brain age model across different age groups within the Singaporean population, encompassing both elderly individuals aged 55 to 88 years and children aged 4 to 11 years. The model, originally trained on a dataset primarily consisting of Caucasian adults, demonstrated a varying degree of adaptability across these age groups. For the elderly, the authors observed that the model could be applied with minimal modifications, whereas for children, significant fine-tuning was necessary to achieve accurate predictions. Through their analysis, the authors established a correlation between changes in the brain age gap and future executive function performance across both demographics. Additionally, they identified distinct neuroanatomical predictors for brain age in each group: lateral ventricles and frontal areas were key in elderly participants, while white matter and posterior brain regions played a crucial role in children. These findings underscore the authors' conclusion that brain age models hold the potential for generalization across diverse populations, further emphasizing the significance of brain age progression as an indicator of cognitive development and aging processes.Strengths:(1) The study tackles a crucial research gap by exploring the adaptability of a brain age model across Asian demographics (Chinese, Malay, and Indian Singaporeans), enriching our knowledge of brain aging beyond Western populations.(2) It uncovers distinct anatomical predictors of brain aging between elderly and younger individuals, highlighting a significant finding in the understanding of age-related changes and ethnic differences.Weaknesses:(1) Clarity in describing the fine-tuning process is essential for improved comprehension.(2) The analysis often limits its findings to p-values, omitting the effect sizes crucial for understanding the relationship with cognition.(3) Employing a predictive framework for cognition using brain age could offer more insight than mere statistical correlations.(4) Expanding the study's scope to evaluate the model's generalisability to unseen Caucasian samples is vital for establishing a comparative baseline.In summary, this paper underscores the critical need to include diverse ethnicities in model testing and estimation.
**Reviewer #1 (Recommendations for the authors):**
Comment #1 - Fine-Tuning Process Clarity: Enhanced clarity in the fine-tuning process documentation is crucial for understanding how models are adapted to new datasets. This involves explaining parameter adjustments and choices, which facilitates replication and application in further research.

We thank Reviewer #1 for this pertinent point. As advised, we have added a Supplementary Methods section with more details on the finetuning process. This includes the addition of Supplementary Figure S6, which shows examples of learning curves that helped inform our parameter adjustments and choices. We have added a reference to this section in *Section 5.2* of the *Methods*.

Comment #2 - Effect Sizes Reporting: The emphasis on reporting effect sizes alongside p-values addresses the need to quantify the strength of observed effects, particularly the relationship between brain age and cognition. Effect sizes provide insights into the practical significance of findings, crucial for clinical and practical applications.

We thank Reviewer #1 for raising this important comment. As suggested, we have added standardized regression coefficients (as measures of effect size) alongside p-values in Figures 3 – 4, Supplementary Figures S2 – S4, Supplementary Tables S4 – S15, and the text of Sections 2.2 – 2.3 of the Results. We have additionally added 95% confidence intervals to Supplementary Tables S4 – S15.

Comment #3 - Predictive Framework for Cognition: Adopting a predictive framework for cognition using brain age moves the research from mere correlation to actionable prediction, offering potentials based on predictive analytics.

We thank Reviewer #1 for this insightful suggestion. Adopting a predictive framework would certainly be a useful and exciting avenue for the application of brain age. However, we note that the current study was primarily interested in the generalizability and interpretability of brain age in Asian children and older adults, as well as the added value of longitudinal measures of brain age. Thus, we believe our correlation-based analysis effectively demonstrated that deviations of brain age from chronological age were not merely random errors, but were informative of cognition. Furthermore, ongoing changes to these deviations were informative of future cognition. This helps to establish the brain age gap as a biomarker for aging, independent of chronological age. Additionally, we expect that the accurate prediction of future cognition would require a multitude of factors, in addition to T1-based brain age, as well as a large sample size to train and test. We believe such a dataset would be a promising avenue for future work, but it is outside the scope of the current study.

Nonetheless, we were able to conduct a preliminary analysis using the current longitudinal data from SLABS and GUSTO. We extracted the same variables used in the original analyses of future cognition, corresponding to Figures 3D and 4B in the main text. To implement a predictive framework, we split the data into 10 stratified cross-validation folds. We also used kernel ridge regression (KRR) as the predictive model, as it has previously shown promising performance in behavioral and cognitive prediction [1]. We used a cosine kernel and nested 5-fold cross-validation to pick the optimal regularization strength (alpha).

To investigate the added value of BAG and longitudinal changes in BAG, we compared 3 predictive models for each cognitive domain. The baseline model consisted of the demographic covariates used in the original analyses (i.e. chronological age, sex, and years of education for older adults). A second model combined demographics with baseline BAG, and the third model incorporated demographics, baseline BAG, and the (early) annual rate of change in BAG. Predictions were extracted from each test fold, and performance was measured by the correlation between test predictions and actual values of future cognition (or change in cognition). Models were statistically compared using the corrected resampled t-test for machine learning models [1], [2], [3]. The Benjamini-Hochberg procedure was used to correct for multiple comparisons.

Author response image 1 shows the prediction results for SLABS and GUSTO. Notably, adding the early change in BAG significantly improves the prediction of future change in executive function in SLABS. There is also an improvement in predicting the future inhibition score in GUSTO, but this is not significant after multiple comparison correction. Encouragingly, these are the same domains that showed significant associations with the change in BAG in the original analyses. This suggests that longitudinal brain age continues to contribute information, independent of baseline factors, in a predictive framework. We hope that future work can expand on this analysis with, for instance, larger sample sizes, more varied and informative predictors, and state-of-the-art prediction methods, in order to establish actionable predictions of future cognition.

**Author response Image 1.**

Predictive framework for cognition similarly suggests value of longitudinal change in BAG. Prediction performance (Pearson's correlation) of KRR across future cognitive outcomes. Each boxplot shows the distribution of performance over cross-validation folds. Model performances are statistically compared for each outcome. Significant outcomes from the original analyses are bolded. (A) Results for SLABS using the early change in BAG and future change in cognitive scores (non-overlapping). Early change in BAG again shows benefit for predicting future change in executive function. (B) Results for GUSTO using the early change in BAG (from 4.5-7.5 years old) and future cognitive score (at 8.5 years old). Early change in BAG again shows benefit for predicting future inhibition, but it is not significant after multiple comparison correction. Key - **: p < 0.01; * (ns): p < 0.05 but p_corr_ > 0.05 after multiple comparison correction; ns: p > 0.05

**Author response image 1. sa2fig1:** 

Comment #4 - Generalizability to Unseen Caucasian Samples: Evaluating the model's performance on unseen (longitudinal) Caucasian samples is important for benchmarking.

We thank Reviewer #1 for this important comment. We agree that generalizability should be benchmarked against performance on unseen Caucasian samples. In the SFCN model paper [4], they conducted an out-of-sample test on unseen Caucasian samples from ages 13 to 95. In this age range, they reported a high correlation (r = 0.975) and low MAE (MAE = 3.90). This favorable generalization performance was verified in adults by independent evaluations [5], [6]. This is also in line with what we observed in Asian older adults, taking into account the different age ranges and sample sizes involved [7].

However, this also highlights the difficulty in evaluating on younger ages in the range of GUSTO (4.5 – 10.5 years old). Most accessible developmental datasets (e.g. HBN, PING) were already included in model training, preventing an unbiased evaluation on these samples. Datasets such as PNC and ABCD were not included in training, but they primarily consist of an older age range than GUSTO. Holm et al. [8] previously tested the SFCN model in ABCD and reported satisfactory performance (low MAE) from 9 – 13 years old. However, to the best of our knowledge, there are no reported generalization results (for any ethnicity) from 4.5 – 7.5 years old, which is where we found the most performance degradation in GUSTO. We are also not aware of any datasets in this age range we could access to test this, unfortunately, but it would be an important area for future work.

While benchmarking in Caucasian children is difficult, we were able to conduct a preliminary analysis with older adults using the ADNI dataset (which was not included in the model training [4]). We selected a longitudinal subset with cognitive data available and no dementia at baseline (N = 137). We used composite cognitive scores covering memory, executive function, language, and visuospatial function [9], [10], [11]. We followed the same methodology (e.g. preprocessing, finetuning, statistical analysis) as the main analyses on EDIS, SLABS, and GUSTO. To maximize the data available, we tested associations with future cognition (taken at the last available time point), similar to GUSTO. We again included chronological age, sex, and years of education as demographic covariates.

Author response image 2 shows the brain age predictions for the pretrained and finetuned models on ADNI. Similar to Singaporean older adults, the pretrained model performs well, producing a high correlation (r = 0.8053; compared to r = 0.7389 for EDIS and r = 0.8136 for SLABS) and somewhat low MAE (MAE = 4.9735; compared to MAE = 3.9895 for EDIS and MAE = 3.4668 for SLABS). After finetuning, the MAE improves (MAE = 3.6837; compared to MAE = 3.3232 for EDIS and MAE = 3.2653 for SLABS) with a similar correlation (r = 0.7854; compared to *r* = 0.7445 for EDIS and *r* = 0.8138 for SLABS). This suggests that generalization to unseen Singaporean older adults is in line with the generalization to unseen Caucasian older adults.

**Author response image 2. sa2fig2:** Brain age predictions on unseen Caucasian sample of older adults. Predictions from the (A) pretrained and (B) finetuned brain age models on ADNI participants. Compare to Figure 2 of the main text.

For the associations with future cognition, we again find that baseline BAG does not associate with future cognition (Author response tables 1 and 2). However, encouragingly, we find that the early annual rate of change in BAG does associate with future memory, which is significant after multiple comparison correction for the finetuned model (Author response tables 2 and 3). This suggests a degree of replicability to the original results, but interestingly, in a different domain (memory vs. executive function). In contrast to SLABS, which consists of healthy older adults recruited from the community, ADNI consists of participants at risk of AD recruited from memory clinics. Thus, this difference in domain could be due to factors such as a stronger signal for memory in the testing battery or greater variations in memory function and decline. However, it could also reflect other population differences between ADNI and SLABS. This is an intriguing area for future study, ideally with larger sample sizes and more diverse populations included.

**Author response table 1. sa2table1:** Linear relationship between pretrained baseline BAG and future cognitive score in ADNI. Compare to Supplementary Tables S4 – S15 of the original text.

Variable of Interest	Outcome	β (CI_L_, CI_U_)	p	P_corr_	Δ R^2^_adj_	R^2^
Baseline BAG (pretrained)	Future Memory	-0.1744 (-0.36, 0.01)	0.0632	0.2442	0.0181	0.1278
Baseline BAG (pretrained)	Future Executive Function	-0.1368 (-0.32, 0.05)	0.1522	0.3043	0.0081	0.0914
Baseline BAG (pretrained)	Future Language	-0.1758 (-0.36, 0.01)	0.0611	0.2442	0.0185	0.1277
Baseline BAG (pretrained)	Future Visuospatial Function	-0.0284 (-0.26, 0.20)	0.8053	0.8053	-0.0098	0.0559

**Author response table 2. sa2table2:** Linear relationship between finetuned baseline BAG and future cognitive score in ADNI. Compare to Supplementary Tables S4 – S15 of the original text.

Variable of Interest	Outcome	β (CI_L_, CI_U_)	p	P_corr_	Δ R^2^_adj_	R^2^
Baseline BAG (finetuned)	Future Memory	-0.1260 (-0.33, 0.07)	0.2155	0.8621	0.0040	0.1142
Baseline BAG (finetuned)	Future Executive Function	0.0244 (-0.18, 0.23)	0.8135	1.0000	-0.0072	0.0766
Baseline BAG (finetuned)	Future Language	-0.0823 (-0.28, 0.12)	0.4193	1.0000	-0.0025	0.1073
Baseline BAG (finetuned)	Future Visuospatial Function	0.0536 (-0.19, 0.30)	0.6648	1.0000	-0.0085	0.0572

**Author response table 3. sa2table3:** Linear relationship between pretrained change in BAG and future cognitive score in ADNI. Compare to Supplementary Tables S4 – S15 of the original text.

Variable of Interest	Outcome	β (CI_L_, CI_U_)	p	P_corr_	Δ R^2^_adj_	R^2^
Change in BAG (pretrained)	Future Memory	-0.1743 (-0.35, -0.00)	0.0492	0.1968	0.0209	0.1549
Change in BAG (pretrained)	Future Executive Function	-0.0711 (-0.25, 0.11)	0.4353	0.8705	-0.0029	0.0959
Change in BAG (pretrained)	Future Language	-0.0577 (-0.23, 0.12)	0.5180	0.8705	-0.0042	0.1306
Change in BAG (pretrained)	Future Visuospatial Function	-0.1337 (-0.35, 0.08)	0.2126	0.6377	0.0061	0.0718

**Author response table 4. sa2table4:** Linear relationship between finetuned change in BAG and future cognitive score in ADNI. Compare to Supplementary Tables S4 – S15 of the original text.

Variable of Interest	Outcome	β (CI_L_, CI_U_)	p	p_corr_	Δ R^2^_adj_	R^2^
Change in BAG (finetuned)	Future Memory	-0.2263 (-0.40, -0.05)	0.0113	0.0451	0.0400	0.1674
Change in BAG (finetuned)	Future Executive Function	-0.1453 (-0.33, 0.04)	0.1153	0.2305	0.0117	0.0966
Change in BAG (finetuned)	Future Language	-0.1737 (-0.35, 0.00)	0.0543	0.1628	0.0204	0.1403
Change in BAG (finetuned)	Future Visuospatial Function	-0.1290 (-0.34, 0.08)	0.2244	0.2305	0.0053	0.0722

References

(1) L. Q. R. Ooi *et al.*, “Comparison of individualized behavioral predictions across anatomical, diffusion and functional connectivity MRI,” *NeuroImage*, vol. 263, p. 119636, Nov. 2022, doi: 10.1016/j.neuroimage.2022.119636.

(2) C. Nadeau and Y. Bengio, “Inference for the Generalization Error,” *Mach. Learn.*, vol. 52, no. 3, pp. 239–281, Sep. 2003, doi: 10.1023/A:1024068626366.

(3) R. R. Bouckaert and E. Frank, “Evaluating the Replicability of Significance Tests for Comparing Learning Algorithms,” in *Advances in Knowledge Discovery and Data Mining*, H. Dai, R. Srikant, and C. Zhang, Eds., Berlin, Heidelberg: Springer, 2004, pp. 3–12. doi: 10.1007/978-3-540-24775-3_3.

(4) E. H. Leonardsen *et al.*, “Deep neural networks learn general and clinically relevant representations of the ageing brain,” *NeuroImage*, vol. 256, p. 119210, Aug. 2022, doi: 10.1016/j.neuroimage.2022.119210.

(5) R. P. Dörfel *et al.*, “Prediction of brain age using structural magnetic resonance imaging: A comparison of accuracy and test-retest reliability of publicly available software packages,” Neuroscience, preprint, Jan. 2023. doi: 10.1101/2023.01.26.525514.

(6) J. L. Hanson, D. J. Adkins, E. Bacas, and P. Zhou, “Examining the reliability of brain age algorithms under varying degrees of participant motion,” *Brain Inform.*, vol. 11, no. 1, p. 9, Apr. 2024, doi: 10.1186/s40708-024-00223-0.

(7) A.-M. G. de Lange *et al.*, “Mind the gap: Performance metric evaluation in brain-age prediction,” *Hum. Brain Mapp.*, vol. 43, no. 10, pp. 3113–3129, Jul. 2022, doi: 10.1002/hbm.25837.

(8) M. C. Holm *et al.*, “Linking brain maturation and puberty during early adolescence using longitudinal brain age prediction in the ABCD cohort,” *Dev. Cogn. Neurosci.*, vol. 60, p. 101220, Feb. 2023, doi: 10.1016/j.dcn.2023.101220.

(9) P. K. Crane *et al.*, “Development and assessment of a composite score for memory in the Alzheimer’s Disease Neuroimaging Initiative (ADNI),” *Brain Imaging Behav.*, vol. 6, no. 4, pp. 502–516, Dec. 2012, doi: 10.1007/s11682-012-9186-z.

(10) L. E. Gibbons *et al.*, “A composite score for executive functioning, validated in Alzheimer’s Disease Neuroimaging Initiative (ADNI) participants with baseline mild cognitive impairment,” *Brain Imaging Behav.*, vol. 6, no. 4, pp. 517–527, Dec. 2012, doi: 10.1007/s11682-012-9176-1.

(11) S.-E. Choi *et al.*, “Development and validation of language and visuospatial composite scores in ADNI,” *Alzheimers Dement. Transl. Res. Clin. Interv.*, vol. 6, no. 1, p. e12072, 2020, doi: 10.1002/trc2.12072.